# Deciphering early human pancreas development at the single-cell level

Zhuo Ma [1,2,11], Xiaofei Zhang[1,3,4,11], Wen Zhong [5,6,11], Hongyan Yi[4], Xiaowei Chen[7], Yinsuo Zhao[1], Yanlin Ma[4,12] ✉, Eli Song [1,12] ✉ & Tao Xu [1,2,8,9,10,12] ✉

Understanding pancreas development can provide clues for better treatments of pancreatic diseases. However, the molecular heterogeneity and developmental trajectory of the early human pancreas are poorly explored. Here, we performed large-scale single-cell RNA sequencing and single-cell assay for transposase accessible chromatin sequencing of human embryonic pancreas tissue obtained from first-trimester embryos. We unraveled the molecular heterogeneity, developmental trajectories and regulatory networks of the major cell types. The results reveal that dorsal pancreatic multipotent cells in humans exhibit different gene expression patterns than ventral multipotent cells. Pancreato-biliary progenitors that generate ventral multipotent cells in humans were identified. Notch and MAPK signals from mesenchymal cells regulate the differentiation of multipotent cells into trunk and duct cells. Notably, we identified endocrine progenitor subclusters with different differentiation potentials. Although the developmental trajectories are largely conserved between humans and mice, some distinct gene expression patterns have also been identified. Overall, we provide a comprehensive landscape of early human pancreas development to understand its lineage transitions and molecular complexity.

The pancreas is an essential digestive and endocrine organ responsible for nutrient metabolism in the body[1,2]. Ninety-nine percent of the pancreatic epithelium consists of exocrine tissue, including the acinar cells that secrete digestive enzymes and the ductal cells that transport these enzymes to the intestine. The remaining 1% consists of endocrine tissue, known as the islets of Langerhans, which comprise five distinct endocrine cell types, including α/β/δ/PP/ε cells, mostly responsible for regulating glucose homeostasis[1]. Dysfunctions of these cells cause a variety of disorders, such as pancreatitis, pancreatic cancer and diabetes. Understanding the embryonic development of the pancreas,

[1]National Laboratory of Biomacromolecules, CAS Center for Excellence in Biomacromolecules, Institute of Biophysics, Chinese Academy of Sciences, Beijing 100101, China. [2]College of Life Sciences, University of Chinese Academy of Sciences, Beijing 100049, China. [3]Key Laboratory of Molecular Biophysics of the Ministry of Education, College of Life Science and Technology, Huazhong University of Science and Technology, Wuhan 430074, China. [4]Hainan Provincial Key Laboratory for Human Reproductive Medicine and Genetic Research, Key Laboratory of Reproductive Health Diseases Research and Translation (Hainan Medical University), Ministry of Education, The First Affiliated Hospital of Hainan Medical University, Hainan Medical University, Haikou 570102, China. [5]Science for Life Laboratory, Department of Biomedical and Clinical Sciences (BKV), Linköping University, Linköping 581 83, Sweden. [6]Department of Neuroscience, Karolinska Institutet, Stockholm, Sweden. [7]Center for High Throughput Sequencing, Core Facility for Protein Research, Key Laboratory of RNA Biology, Institute of Biophysics, Chinese Academy of Sciences, Beijing 100101, China. [8]Guangzhou Laboratory, Guangzhou 510005, China. [9]Central Hospital Affiliated to Shandong First Medical University, Jinan 250013, China. [10]Medical Science and Technology Innovation Center, Shandong First Medical University & Shandong Academy of Medical Sciences, Jinan 250062, China. [11]These authors contributed equally: Zhuo Ma, Xiaofei Zhang, and Wen Zhong. [12]These authors jointly supervised this work: Yanlin Ma, Eli Song, and Tao Xu. ✉e-mail: mayanlinma@hotmail.com; songali@ibp.ac.cn; xutao@ibp.ac.cn

especially cell fate decisions and endocrine cell differentiation, may help us to improve the differentiation protocols for pancreatic cells from human pluripotent stem cells (hPSCs) in vitro[3–9].

Benefitting from powerful genetic tools and animal models, previous studies have revealed the important molecular events for pancreas organogenesis in rodents[1,10,11]. The pancreas originates from both the dorsal and ventral endoderm domains, which receive different signals from adjacent tissues. This process begins on embryonic day (E) 8.5 in mice and at 27-29 days post conception in humans[12,13]. The ventral pancreas emerges later than the dorsal pancreas, and it shares common progenitors with the liver and extrahepatic bile ducts (EHBD)[14,15]. The two pancreatic buds contain multipotent progenitor (MP) cells that can differentiate into all lineages of the pancreatic epithelium and eventually fuse to form a single organ due to gut tube rotation. PDX1, FOXA2 and PTF1A are the key regulators of MP cell specification[16,17]. Cell proliferation enlarges the pancreatic epithelium and reshapes it to form branched tubular structures. Simultaneously, MP cells develop into tip cells that have the potential to differentiate into acinar cells and trunk cells that have the bipotential to differentiate into endocrine and ductal cells[18,19]. Two transcription factors (TFs), PTF1A which promotes tip fate and NKX6-1 which induces trunk fate are the master regulators in this process[19]. Endocrine progenitor (EP) cells are regulated by NEUROG3 and delaminate from the trunk domain to differentiate into endocrine cells[20,21]. Beta cell maturation, which is regulated by certain factors such as MAFA, is a long process[22,23]. While many conclusions about pancreas development and in vitro differentiation protocols have been made based on the studies in mice, species differences between humans and mice have been noticed[24,25]; for example, there are two peaks of endocrine differentiation in mice and only a single phase in humans[13,26].

Single-cell RNA sequencing (scRNA-seq) and single-cell assay for transposase accessible chromatin sequencing (scATAC-seq) are powerful tools that have already been applied in developmental biology. Recent studies in mice and humans have revealed the cellular composition, molecular heterogeneity and developmental trajectory of pancreatic cells in fetuses at the single-cell resolution[27–36]. However, due to the scarcity of human embryo samples and the difficulty of pancreas isolation from early embryos, little is known about the molecular features and regulatory network of early pancreatic development in humans, especially before post-conception week (PCW) 8.

Here, we performed scRNA-seq of human embryonic pancreas tissues collected from PCW 4 to 11 and scATAC-seq of tissues from PCW 8 to 11. We profiled the major cell types of pancreatic epithelial cells, including dorsal and ventral MP cells, and revealed their molecular heterogeneity and developmental trajectories. We analyzed the TFs, regulatory networks and signaling pathways of acinar and ductal lineage cells and revealed the regulatory network of endocrinogenesis and the transcription dynamics of EP and endocrine cells. We further compared the developmental trajectories between humans and mice and identified several distinct features in each species, such as differences in gene expression patterns. Taken together, our data depict the whole trajectory of pancreatic organogenesis during the first trimester (PCW 4-11) at the single-cell level.

## Results
### Cell diversity of the human pancreas in early development
We collected human embryonic pancreas samples at 8 time points from PCW 4 to 11 from 17 donors, including 6 males and 11 females (Supplementary Data 1). After the digestion of the isolated pancreas, we performed scRNA-seq of all 17 processed samples using the 10x Genomics platform (Fig. 1a, Supplementary Data 1). In total, 68,714 cells passed the quality control procedures, with an average of 3,000 expressed genes per cell (Supplementary Fig. 1a). Our data showed high similarity among the samples from the same time point

(Supplementary Fig. 1b, c). Because PCW 4-6 samples also contained non-pancreatic cells, we analyzed and presented our dataset in two groups for batch correction, dimension reduction and clustering. A total of six major cell-type classes were identified, including epithelial (EPCAM + ), mesenchymal (COL3A1 + ), endothelial (PECAM1 + ), neural (ASCL1 + ), immune (PTPRC + ), and erythroid (HBA1 + ) cells (Supplementary Fig. 1d–h). Our data showed the continuity of cells in the same cell class across different time points (Supplementary Fig. 1i). Mesenchymal cells constituted the majority of both two datasets, and their proportion differed more obviously between PCW 4-6 and PCW 7-11 owing to the different sample isolation methods (Supplementary Fig. 1h, i). From PCW 7 to 11, the proportion of mesenchymal cells gradually decreased, while epithelial and other classes of cells increased (Supplementary Fig. 1i).

### Molecular heterogeneity in the early human embryonic pancreatic epithelium
Epithelial cells are the main components of the pancreas and perform its basic functions of the pancreas. To investigate the transcriptional profile of the development of pancreatic epithelial cells, we used EPCAM and PDX1 as pancreatic epithelial markers. Based on this approach, we obtained a total of 17,135 pancreatic epithelial cells from PCW 4 to 11 and included all of them into a common cluster analysis. We identified 13 single-cell-type clusters, which represented all epithelial lineages in the early development of the pancreas (Fig. 1b, c, Supplementary Fig. 2a, b). MP cells only existed in PCW 4 to 5, during which their numbers gradually decreased (Fig. 1c). The numbers of early tip and early trunk cells gradually increased from PCW 4 to 7. Duct and acinar cells emerged after PCW 10 (Fig. 1c). A large number of EPs and endocrine cells were generated after PCW 8 (Fig. 1c). The expression patterns of the top 100 differentially expressed genes (DEGs) in each cluster were visualized in a heatmap, with TFs highlighted (Fig. 1d, Supplementary Fig. 2d and Supplementary Data 2). GATA4- and FOXA2-positive MP cells could be divided into two groups; the dorsal MP cells expressed NR2F1, while the ventral MP cells expressed TBX3 (Fig. 1d, e, Supplementary Fig. 2d, e). We present the analyses of the dorsal and ventral MP cells in later sections. Early tip, tip, and acinar cells expressed increasing levels of CPA2, RBPJL and CTRB2 (Fig. 1e, Supplementary Fig. 2d, e). The acinar cells expressing CLPS and CTRB1, which mainly appeared in PCW 11, did not show the expression of any amylase-associated genes (Fig. 1c, Supplementary Fig. 2d, e). These results indicated that the acinar cells identified in our dataset during embryonic development were still immature and lacked sufficient digestive enzymes. Early trunk, trunk, and duct cells expressed increasing levels of HES4, DCDC2, and CFTR (Fig. 1d, e, Supplementary Fig. 2d, e). Several endocrine cells expressed high levels of endocrine hormones, and EP cells expressed high levels of NEUROG3 (Fig. 1d, Supplementary Fig. 2d, e). A negligible number of endocrine cells were found in PCW 4 samples, and these cells disappeared in PCW 5-7 samples (Fig. 1c, Supplementary Fig. 2b, c). Then, the population of endocrine cells gradually increased in pancreatic epithelial cells after PCW 8 and peaked at PCW 10 and 11 (Fig. 1c, Supplementary Fig. 2b, c), suggesting that large numbers of endocrine cells differentiated after PCW 8. The developmental trajectory constructed by Monocle3[37] showed the branches of acinar, duct, and endocrine lineage cells (Fig. 1f). Considering these results combination with RNA velocity analysis[38,39], we inferred that dorsal and ventral MP cells differentiated into early tip and trunk cells. Then, the early tip cells developed into tip and acinar cells, while the early trunk cells developed into trunk and duct cells. Trunk cells are bipotential cells that differentiate into EP and duct cells (Supplementary Fig. 2f). Collectively, these results showed the developmental trajectory of pancreatic epithelial cells in PCW 4-11. Next, we focused on the development of major lineages in pancreatic epithelial cells.

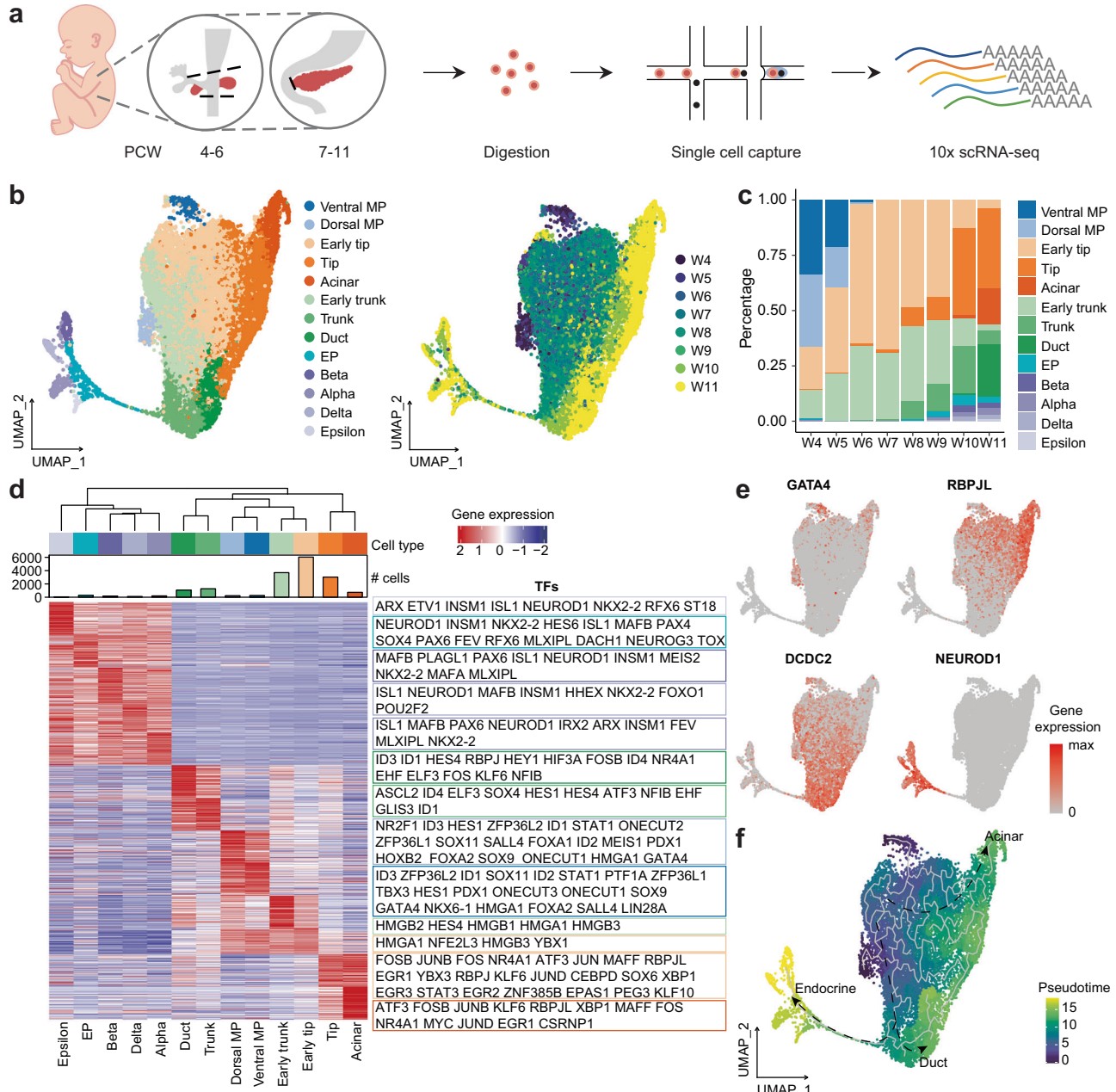

**Fig. 1 | scRNA-seq identified major cell types in the early human embryonic pancreatic epithelium. a** Schematic diagram of the procedures for sample information, tissue processing, and scRNA-seq profiling methods. PCW, post-conception week. **b** UMAP plot of all single cells colored by cell type and time point in pancreatic epithelial cells. UMAP uniform manifold approximation and projection, MP, multipotent progenitor, EP endocrine progenitor. **c** Bar plot showing the percentage of each cell type in pancreatic epithelial cells. **d** Heatmap showing the scaled expression of the top 100 differentially expressed genes in pancreatic epithelial cells. Transcription factors of each cell type are labeled on the right. **e** Feature plot showing the expression of key marker genes of pancreatic epithelial cells. **f** UMAP plot showing the developmental trajectories of pancreatic epithelial cells. See also Supplementary Figs. 1, 2 and Supplementary Data 1, 2.

## Dorsal and ventral multipotent progenitor cells have different expression patterns

The pancreas arises from two buds located on the dorsal and ventral sides of the distal foregut endoderm. Beginning in PCW 6 in humans, the ventral pancreas contacts the dorsal pancreas as a result of gut rotation and finally fuses in PCW 8[40–42]. To characterize the differences between dorsal and ventral MP cells, a differential gene expression analysis was conducted (Fig. 2a). Gene Ontology (GO) analysis of the identified DEGs showed that dorsal MP cells were related to Wnt signaling, cell junction assembly and synapse organization, while ventral MP cells were associated with

ribosome assembly, muscle tissue development and myoblast differentiation (Fig. 2b). Both dorsal and ventral MP cells expressed common pancreatic markers such as *PDX1*, *PTF1A*, and *NKX6-1* (Fig. 2c). However, ventral MP cells specifically expressed the TFs *TBX3* and *SOX6*, while dorsal MP cells expressed *NR2F1* and *SIM1* (Fig. 2a, c). Wnt signaling-related genes (*GPC3*, *FRZB*, *FZD5* and *LYPD6*) were highly expressed in dorsal MP cells, while the BMP signaling target TFs (*ID1*, *ID2* and *ID3*) were highly expressed in ventral MP cells (Fig. 2c). These results implied that dorsal and ventral MP cells received different signals during their development.

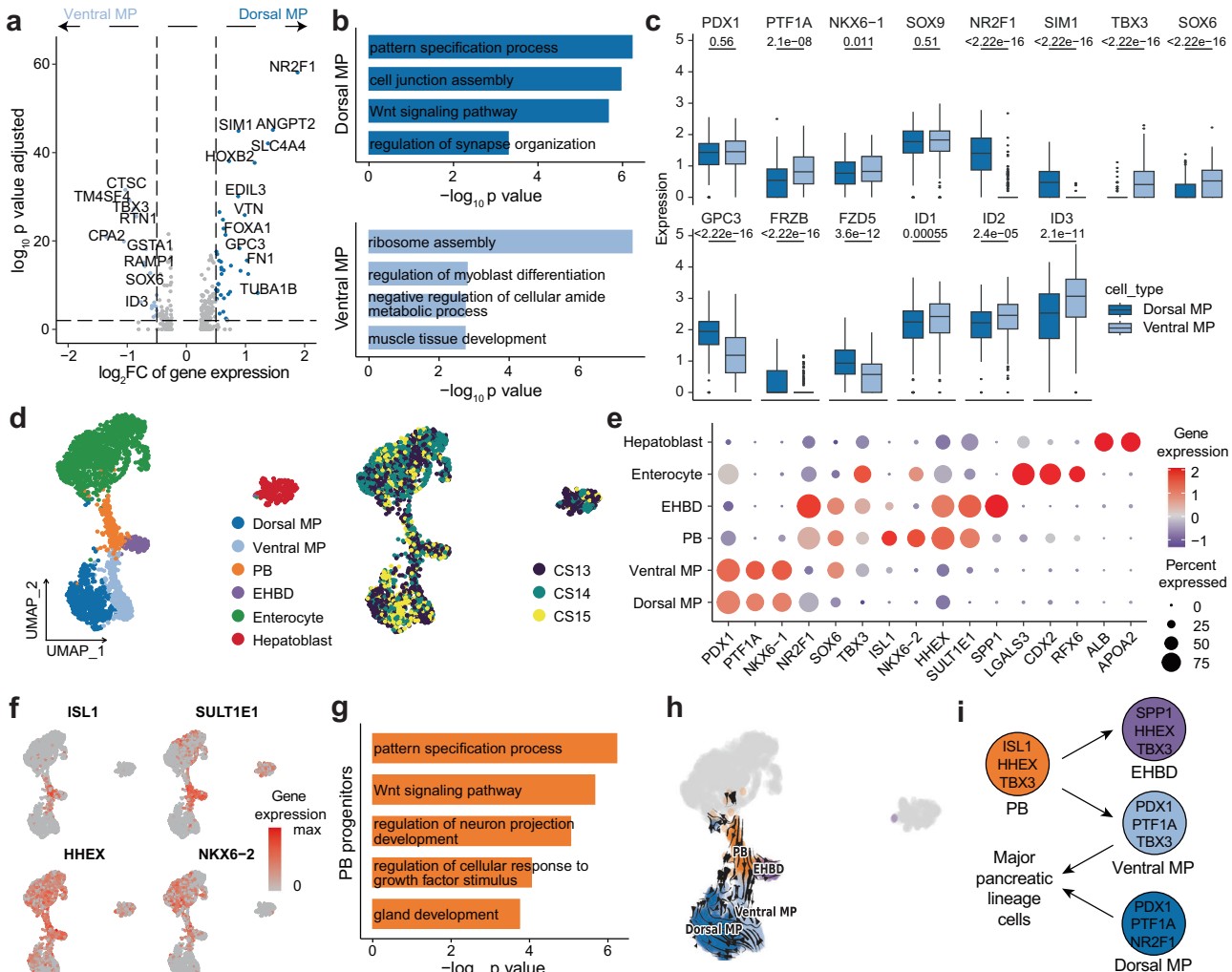

**Fig. 2 | Molecular diversity of epithelial cells in the early pancreas and peri-pancreatic organs. a** Volcano plot showing the differential gene expression between dorsal MP and ventral MP cells. Adjusted *p*-values were calculated by two-sided Wilcoxon test with Bonferroni correction. FC, fold change. **b** Bar plot showing the enriched GO terms of differentially expressed genes in dorsal MP and ventral MP cells. *P*-values were calculated by using *enrichGO* function from R package clusterProfiler with one-sided hypergeometric test. **c** Box plots showing key gene expression in dorsal MP and ventral MP cells. The numbers above the box plots represent the *p*-values calculated using the Wilcoxon test with two-sided comparisons. The center line, bounds of box, whiskers, and single points represent median, 25th to 75th percentile range, 5th and 95th percentile range as well as outliers.

Dorsal MP cells, *n* = 607 cells; Ventral MP cells, *n* = 324 cells. **d** UMAP plot of all single cells colored by cell type and time point in early epithelial cells. CS Carnegie stages, EHBD extrahepatic bile ducts, PB pancreato-biliary progenitors. **e** Dot plot showing cell type marker gene expression in early epithelial cells. **f** Feature plot showing the expression of marker genes of PB progenitors. **g** Bar plot showing the enriched GO terms of differentially expressed genes in PB progenitors. *P*-values were calculated by using *enrichGO* function from R package clusterProfiler with one-sided hypergeometric test. **h** RNA velocity plot showing the developmental trajectory of PB, EHBD, ventral MP, and dorsal MP cells. **i** Model of PB progenitor differentiation. See also Supplementary Fig. 3 and Supplementary Data 3.

## Ventral multipotent progenitor cells originate from pancreato-biliary progenitors

Other endoderm-derived organ primordia, such as the liver, EHBD (including gallbladder) and duodenum, are adjacent to the dorsal and ventral pancreas[14,15]. The hepato-pancreato-biliary organ system originates from a common ventral endoderm progenitor compartment in mice[14]. To further characterize the relationships between these organs, we analyzed the epithelial cells from these organs in PCW 4 to 5. Six cell clusters were identified based on tissue-specific genes, including dorsal MP (*PDX1*+/*NR2F1*+), ventral MP (*PDX1*+/*TBX3*+), pancreato-biliary (PB) progenitors (*ISL1*+/*SULT1E1*+), EHBD (*SPP1*+/*SULT1E1*+), enterocyte (*CDX2*+), and hepatoblast (*ALB*+) (Fig. 2d, e, Supplementary Fig. 3a–c and Supplementary Data 3). We identified a small population of cells with low expression of *PDX1* and high expression of *HHEX, ISL1, SULT1E1*, and *NKX6-2* and defined them as PB progenitors (Fig. 2d–f). GO analysis of DEGs in PB progenitors showed that these

cells were related to pattern specification process, Wnt signaling pathway, neuron projection development, growth factor stimulus and gland development (Fig. 2g). RNA velocity analysis demonstrated that ventral MP and EHBD cells originated from PB progenitors (Fig. 2h). We also depicted the developmental trajectory of PB, EHBD, and ventral MP cells and calculated the pseudotime for each cell type with Monocle3 (Supplementary Fig. 3d)[37]. The results revealed that PB progenitors have the potential to differentiate into ventral MP and EHBD cells. No connectivity between PB progenitors and hepatoblasts was observed. We performed differential gene expression analysis between ventral MP and EHBD cells. Some TFs, such as *PDX1, NKX6-1, RBPJ*, and *PTF1A*, were expressed at high levels in ventral MP cells, while *NR2F2, SOX4, HHEX, DACH1, ONECUT2*, and *NR2F1* were expressed at moderate levels in EHBD cells (Supplementary Fig. 3e). FGF and Slit-Robo signaling were increased in EHBD cells, while Notch signaling was upregulated in ventral MP cells (Supplementary Fig. 3e). Collectively,

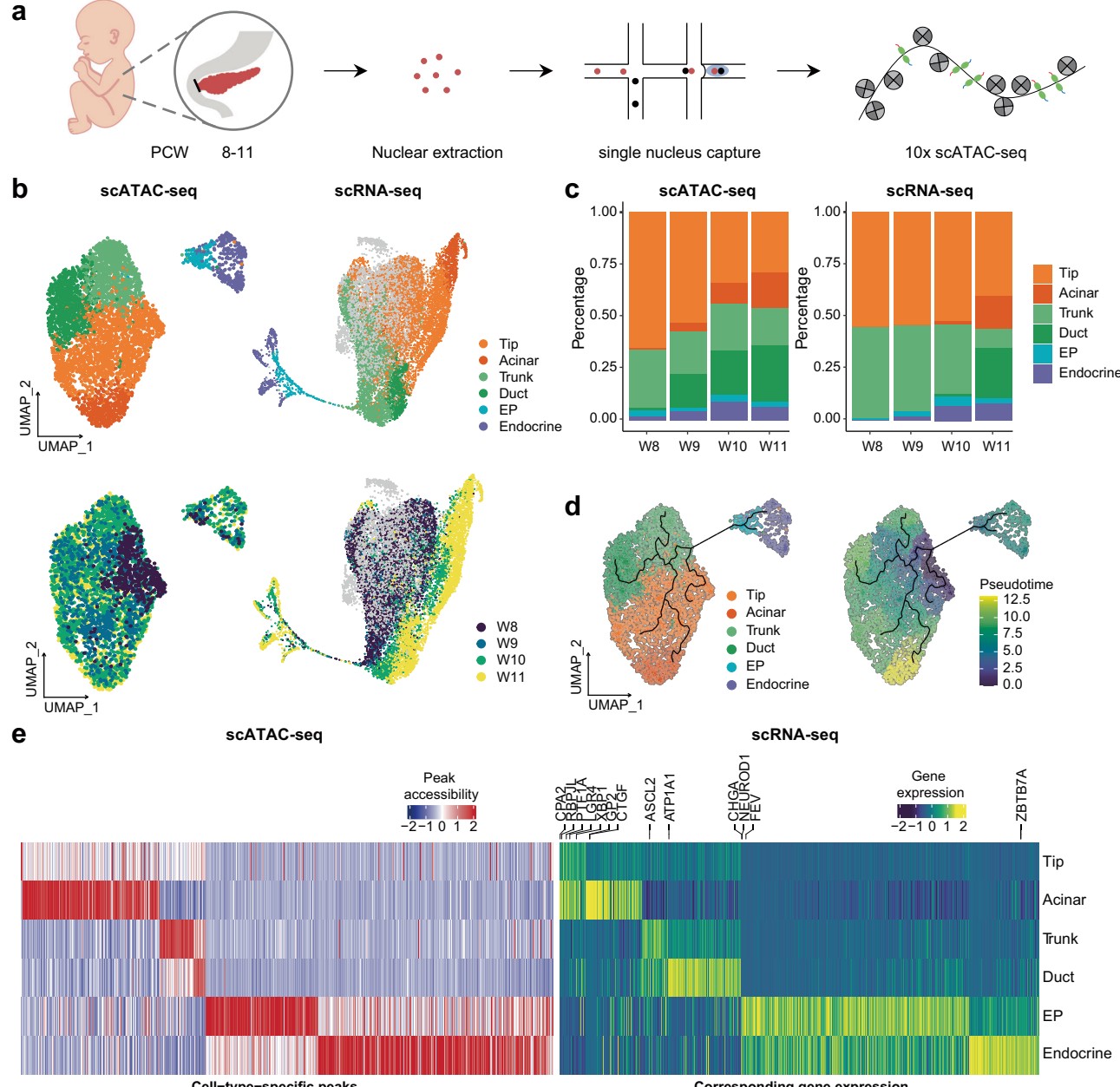

**Fig. 3 | Single-cell chromatin accessibility of pancreatic epithelial cells.**
**a** Schematic diagram of the procedures for sample information, tissue processing, and scATAC-seq profiling methods. **b** UMAP plot of all single cells colored by cell type and time point in pancreatic epithelial cells. **c** Bar plot showing the percentage of each cell type in the scATAC-seq and scRNA-seq. **d** UMAP plot showing the development trajectory of epithelial cells in scATAC-seq. **e** Heatmap showing cell type specific peaks and their corresponding gene expression in scRNA-seq. See also Supplementary Fig. 4 and Supplementary Data 4.

PB progenitors could differentiate into ventral MP and EHBD cells (Fig. 2i).

## scATAC-seq profiling of the developing human pancreas

To investigate the underlying gene regulatory programs driving the bifurcation of cell fate and the continuous differentiation of each pancreatic lineage, we performed scATAC-seq profiling of human embryonic pancreas tissues from PCW 8-11 (Fig. 3a). We obtained 12,288 cells after quality control (Supplementary Fig. 4a). The major cell classes identified by scATAC-seq were consistent with the scRNA-seq results (Supplementary Fig. 4b, c), among which epithelial cells were inspected more closely to focus on pancreatic epithelium differentiation. Cell types in acinar, ductal and endocrine lineages were identified in the scATAC-seq data, consistent with the scRNA-seq

results (Fig. 3b). The chromatin accessibility patterns of marker genes for each cell type were consistent with their cellular expression (Supplementary Fig. 4d). Intriguingly, specific cell populations, including acinar, duct EP and endocrine cells, were identifiable earlier in the scATAC-seq data than in the scRNA-seq; that is, the corresponding marker regions appeared to be accessible earlier than the mRNAs were detected (Fig. 3c, Supplementary Fig. 4e), which may reflect the occurrence of gene regulation prior to gene expression. Pseudotime analysis revealed similar differential trajectories to those observed in scRNA-seq, reflecting high consistency between the scRNA-seq and scATAC-seq profiles (Fig. 3d). The expression patterns of cell-type-specific peaks and their corresponding genes were compared for each cell type (Fig. 3e, Supplementary Data 4). Notably, there were far more distinct peaks in mature acinar cells and endocrine lineage cells,

possibly due to their relative maturity and gradual acquisition of more complex functions (Fig. 3e).

## Cell fate determination in developing acinar and ductal lineage cells

The main populations in the pancreatic epithelium are composed of acinar (tip) and ductal (trunk) lineage cells. To investigate the cell fate determination of these two lineages, we compared their gene expression patterns. *HES4* and *RBPJL* were differentially expressed between early tip and early trunk cells, and the differences were more pronounced between mature tip and trunk cells (Supplementary Fig. 5a, Fig. 4a). We performed branched gene expression analysis on the differential directions of tip and trunk cells and highlighted the TFs (Fig. 4b). GO analysis of these genes revealed that Notch, Wnt, MAPK pathway, actin, cell junctions and nervous system development were activated in trunk and duct cells, whereas the metal ion homeostasis, peptidase activity, digestion and intrinsic apoptotic signaling pathway were activated in tip and acinar cells (Fig. 4c). The active Notch signaling pathway and its downstream TF Hes1 make MP cells acquire a trunk and ductal fate in mice[19,43–45]. We found that *HES1* and another Notch signaling-related TF, *HEY1*, were highly expressed in ductal lineage cells (Fig. 4d). Moreover, ductal lineage cells showed higher expression of *HES4*, which is a homeolog of *HES1* and shows no orthologous genes in mice (Fig. 4d, e). This finding indicated that the Notch signaling pathway is important for ductal lineage cell specification in the human pancreas. To characterize the signaling pathway between tip, trunk cells and supporting cells, we divided supporting cells into six clusters, including fibroblasts, mesothelial cells, pericytes, immune cells, neural cells and endothelial cells (Supplementary Fig. 5b, c). Cell–cell communication analysis identified interactions via the Notch signaling pathway mainly involving DLK1 and JAG1 in fibroblasts and pericytes and NOTCH1, NOTCH2 and NOTCH3 in trunk and duct cells (Fig. 4f, Supplementary Fig. 5e, f). The FGF signaling pathway is important for pancreatic development[46–48]. We found that FGF signaling was much stronger in trunk and duct cells than in tip and acinar cells (Supplementary Fig. 5d). *FGFR2* was highly expressed in trunk and ductal cells, and its ligands *FGF7* and *FGF9* were expressed in fibroblasts and mesothelial cells (Fig. 4g, Supplementary Fig. 5d, f, g). In addition, an NTF4-NTRK2 interaction was identified between mesothelial cells and ductal lineage cells, and a BDNF-NTRK2 interaction was identified between pericytes and ductal lineage cells (Fig. 4g, Supplementary Fig. 5f, g). FGFR2 and NTRK2 are both tyrosine kinase receptors that take part in the MAPK cascade. These results showed that Notch and MAPK signaling promoted trunk and duct cell differentiation. In addition, we found that the HGF signaling pathway was important for acinar lineage cells (Supplementary Fig. 5e). *MET* was specifically expressed in tip and acinar cells, and its ligand *HGF* was expressed in pericytes and mesothelial cells (Fig. 4g, Supplementary Fig. 5f, g).

To construct gene regulatory networks (GRNs) that regulate cell differentiation, we integrated time-matched scATAC-seq and scRNA-seq based on canonical correlation analysis (CCA) and generated GRNs according to IReNA2 with slight modifications[49]. We combined the cell types in each lineage together in scATAC-seq analysis and focused on the inter-relationships between cell lineages. Key regulators and the corresponding regulatory networks driving acinar and ductal lineage specification were identified. We found sets of TFs that were specifically enriched in cell-type-specific peaks in either of these two lineages, as well as a set of TFs that were commonly enriched in both lineages (Fig. 4h). The motif enrichment of lineage-specific TFs was consistent with their expression levels (Fig. 4i). The average expression levels of the three sets of TFs in scRNA-seq were consistent with their footprint enrichment in scATAC-seq (Fig. 4j, Supplementary Fig. 5h). Among the lineage-specific TFs, the Notch signaling-related TFs HES1, HES4 and HEY1 were enriched in the ductal lineage, consistent with their high

expression levels in ductal lineage cells (Fig. 4d). Known key regulators identified in mouse acinar cells, such as XBP1 and ONECUT1, were also enriched in human acinar lineages, as well as a few potential new regulators, such as CEBPD and SOX6. EPAS1 may also play an important role in acinar lineage development, as it targets other key acinar TFs, such as XBP1 and JUNB (Fig. 4i, Supplementary Fig. 5i). *Epas1* was transiently expressed in mouse multipotent pancreatic progenitors and not in differentiated endocrine or exocrine cells[50], while in the developing human pancreas, *EPAS1* was uniquely expressed in acinar cells. Both deficiency and overexpression of Epas1 have been reported to severely impair acinar cell development in mice[50,51], although its role in human acinar development remains to be investigated. The lineage-specific TFs had lineage-specific targets, including markers of either acinar or duct cells, such as *REG4* and *LGR4* for acinar cells or *KRT17*, *KRT19* and *HES4* for duct cells (Supplementary Fig. 5i). For the lineage-common TFs, GRN analysis revealed that despite enriched footprints being observed in both lineages, these TFs had different targets in the two lineages. In acinar lineages, their targets included various kinds of enzymes and protease inhibitors, such as *CELA2A* and *SERPINA4*, and in ductal lineages, their targets were *COL4A1*, *COL4A2* and *KRT17*, which are marker genes of duct cells, and other ductal lineage-specific TFs, such as *HES4* and *HES1* (Fig. 4k, Supplementary Fig. 5i).

## Transcriptional dynamics in bipotent trunk cell development

Trunk cells are bipotent progenitors with the potential to generate both duct cells and endocrine cells, for which the driving force has not been fully deciphered. To address this issue, we performed a branched gene expression analysis on the developmental directions of EP and duct cells with TFs highlighted (Fig. 5a). The GO analysis of cluster-1 genes, including *ASCL2*, indicated that these genes were associated with stem cell differentiation, tight junction assembly and BMP signaling (Fig. 5b). *ASCL2* was highly expressed in trunk and EP cells but was expressed at low levels in duct cells (Fig. 5c, d). These findings indicated that *ASCL2* might be related to endocrinogenesis. Cluster-2 genes, such as *NEUROG3*, *FEV* and *INSM1*, were highly expressed in EP cells and enriched in GO terms for endocrine pancreas development, neurogenesis, peptide secretion and cytoskeleton (Fig. 5a, b). Cluster-3 genes, such as *HES4* and *NFIB*, were upregulated in duct cells and were related to the Notch, Wnt, TGFβ, and ERK pathways and epithelial cell proliferation (Fig. 5a, b). We also found that the histone deacetylase *HDAC2*, which can recruit *KDM1A* and *RCOR2* to form a complex, was upregulated in EP cells (Fig. 5e, Supplementary Fig. 6a), indicating a potential role of epigenetic regulation in cell fate determination. This inference was corroborated in another published human embryonic pancreas dataset (Supplementary Fig. 6b)[28].

To explore the candidate regulators favoring differentiation toward a specific cell lineage, we focused on TFs in trunk cells whose targets were other TFs that were specifically enriched in either duct or endocrine progenitors (details in Methods) (Fig. 5f). In the trunk endocrine transition, *YBX1* upregulated *INSM1*, which is a key regulator in the endocrine lineage and has been reported to participate in mouse islet development and β-cell identity maintenance[52,53]. *INSM1*, in turn, upregulated a series of genes confined to endocrine lineages, including *FEV* and *PAX6*, which are reported to be key regulators in the development of EP cells and to trigger a complex network that sustains EP identity (Fig. 5g and Supplementary Fig. 6c)[27,28,31,34,36]. Coexpression patterns and chromatin accessibility patterns corroborated the GRN prediction of the *YBX1-INSM1-FEV* relationship (Fig. 5h, Supplementary Fig. 6d). Likewise, as targets of *ASCL2*, a decreased level of *ELF3* suppressed duct commitment, and an increased level of *MAFA* favored the development of EP cells (Fig. 5f). On the other hand, *ID4* upregulated *HES4*, and *HES4* downregulated *FOXA2*, which in turn upregulated the known key EP regulator *NEUROD1*[54,55] to suppress endocrine commitment (Fig. 5f, Supplementary Fig. 5g). *HES4* also upregulated *GLIS3* and other TFs in duct cells to favor duct commitment (Fig. 5g).

 

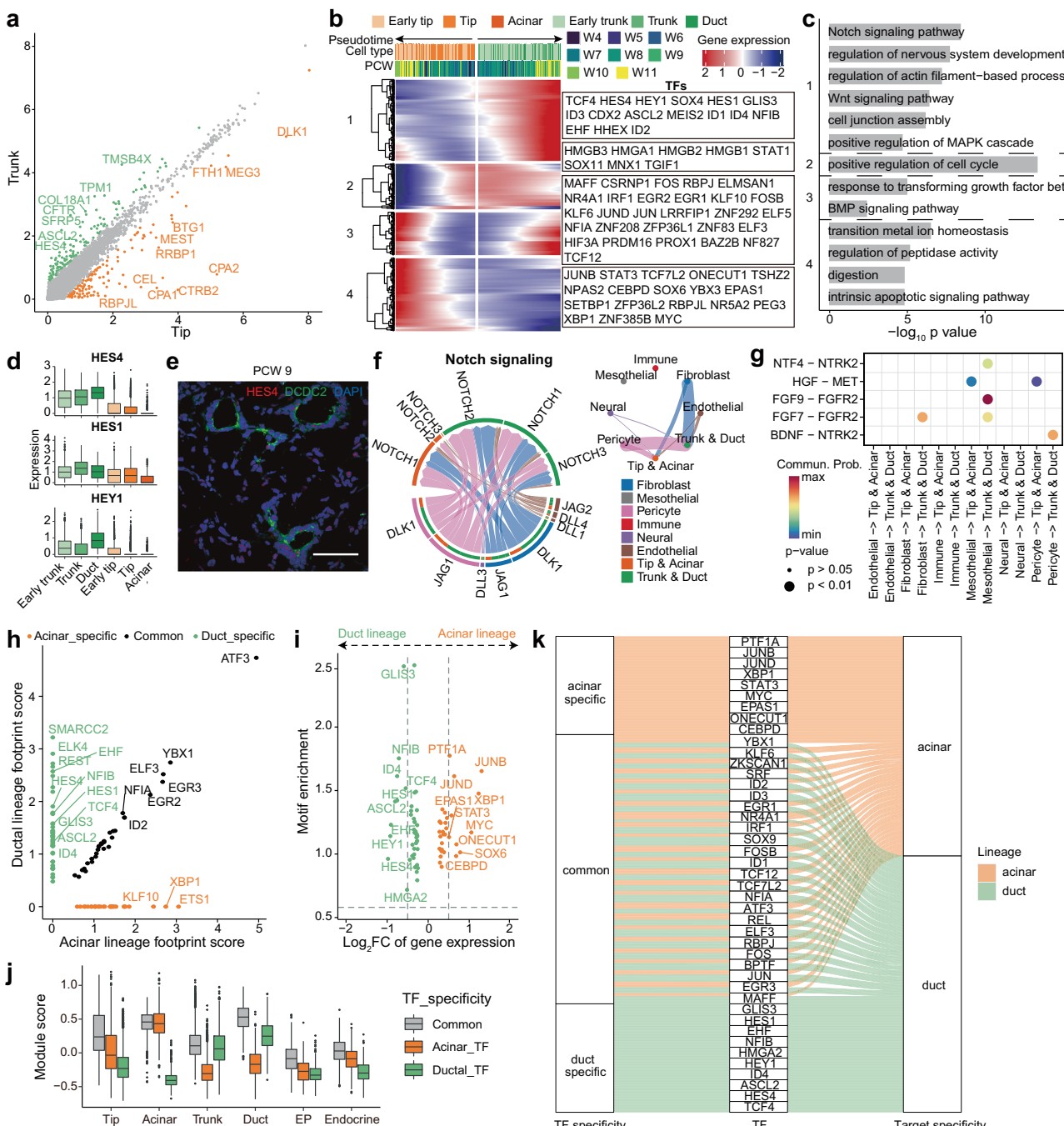

**Fig. 4 | Comparison between acinar and ductal lineage cells during pancreas development. a** Scatter plot showing the mean expression of differentially expressed genes between tip and trunk cells. **b** Heatmap showing the bifurcation of gene expression along the developmental trajectory of ductal lineage and acinar lineage cells. **c** Bar plot showing the enriched GO terms of clustered genes in Fig. 4b. *P*-values were calculated by using *enrichGO* function from R package clusterProfiler with one-sided hypergeometric test. **d** Box plot showing the expression of Notch signaling pathway-related genes. The center line, bounds of box, whiskers, and single points represent median, 25th to 75th percentile range, 5th and 95th percentile range as well as outliers. Early trunk, $n = 3696$ cells; Trunk, $n = 1274$ cells; Duct, $n = 1066$ cells; Early tip, $n = 6033$ cells; Tip, $n = 3029$ cells; Acinar, $n = 731$ cells. **e** Immunostaining of HES4 and DCDC2 in the PCW 9 pancreas. Scale bar, 50 μm. Images shown are representatives of more than three samples from three

independent experiments. **f** Network plot showing Notch signaling pathway-related interactions between acinar, ductal lineage cells and supporting cells. **g** Dot plot showing the communication probability and *p*-value of selected interactions between acinar and ductal lineage cells and supporting cells. *P*-values are calculated from one-sided permutation test. **h** Scatter plot showing the footprint score in acinar and ductal lineages. **i** Scatter plot showing the log2FC of gene expression in scRNA-seq and TF motif enrichment in scATAC-seq for lineage-specific TFs. **j** Average expression of the three sets of TFs in scRNA-seq. The center line, bounds of box, whiskers, and single points represent median, 25th to 75th percentile range, 5th and 95th percentile range as well as outliers. Tip, $n = 5745$ cells; Acinar, $n = 731$ cells; Trunk, $n = 3392$ cells; Duct, $n = 1066$ cells; EP, $n = 274$ cells; Endocrine, $n = 533$ cells **k** Summary of the TF and target specificity of the three sets of TFs. See also Supplementary Fig. 5.

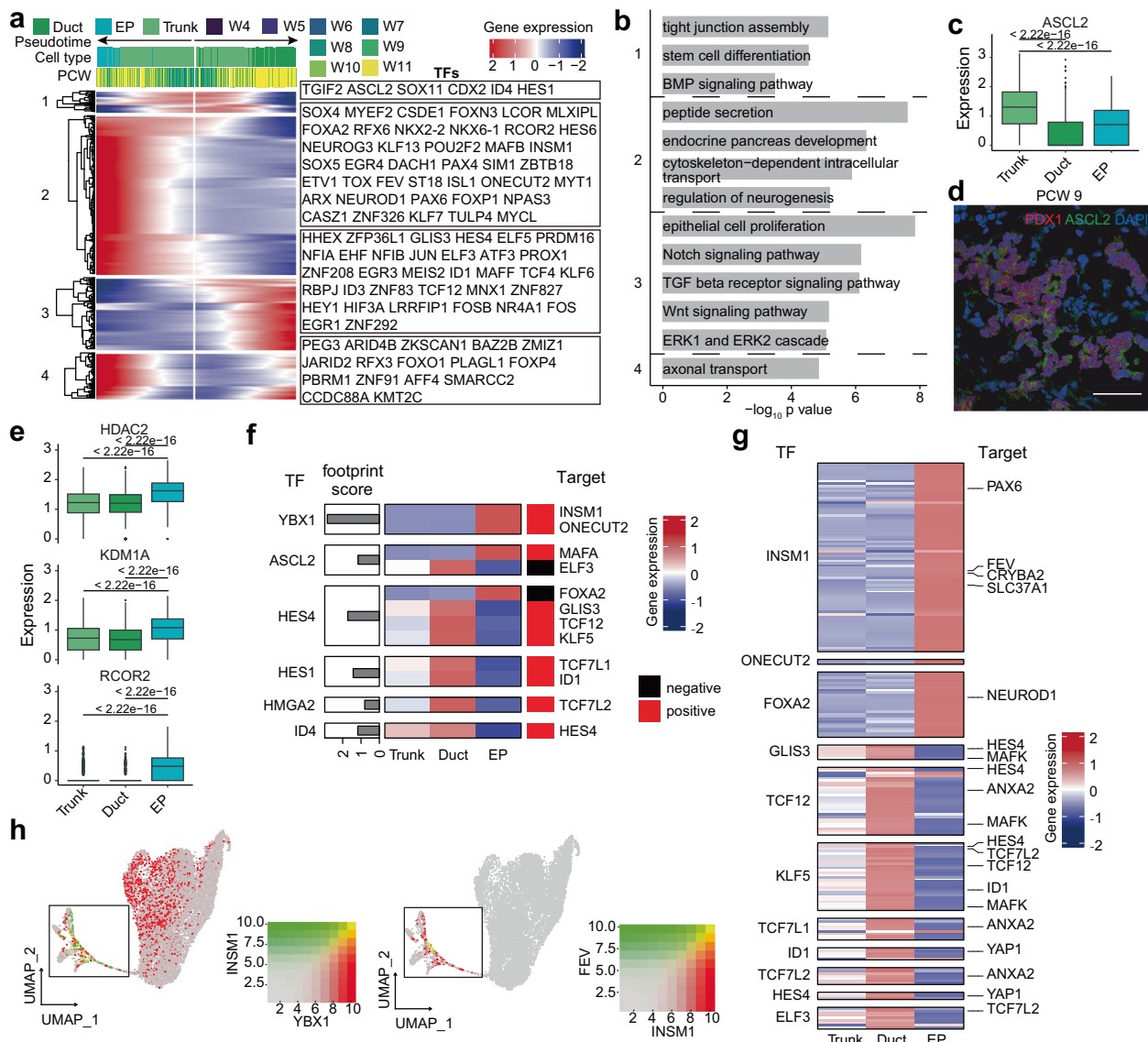

**Fig. 5 | Transcriptional regulation of bipotent trunk cell development.** **a** Heatmap showing the bifurcation of gene expression along the developmental trajectory of duct and EPs. **b** Bar plot showing the enriched GO terms of clustered genes in Fig. 5a. *P*-values were calculated by using *enrichGO* function from R package clusterProfiler with one-sided hypergeometric test. **c** Box plot showing the expression of *ASCL2*. The numbers above the box plots represent the *p*-values calculated using the Wilcoxon test with two-sided comparisons. The center line, bounds of box, whiskers, and single points represent median, 25th to 75th percentile range, 5th and 95th percentile range as well as outliers. Duct, *n* = 1066 cells; EP, *n* = 276 cells; Trunk, *n* = 1274 cells. **d** Immunostaining of ASCL2 in PCW 9

pancreas. Scale bar, 50 μm. Images shown are representatives of more than three samples from three independent experiments. **e** Box plot showing the expression of some epigenetic regulation enzymes. The numbers above the box plots represent the *p*-values calculated using the Wilcoxon test with two-sided comparisons. The center line, bounds of box, whiskers, and single points represent median, 25th to 75th percentile range, 5th and 95th percentile range as well as outliers. Duct, *n* = 1066 cells; EP, *n* = 276 cells; Trunk, *n* = 1274 cells. **f** Heatmap showing the expression of targets of trunk TFs. **g** Heatmap showing the expression of targets of duct and endocrine TFs. **h** Feature plot showing the coexpression levels of selected genes. See also Supplementary Fig. 6.

## Molecular heterogeneity of developing human pancreatic endocrine cells

To further investigate the molecular mechanisms of endocrine development, we performed a clustering analysis of EPs and endocrine cells. A total of five clusters of EP cells expressing *NEUROG3* and four clusters of endocrine cells expressing endocrine hormone genes were identified (Fig. 6a and c, Supplementary Fig. 7b and Supplementary Data 5). Five subclusters of EP cells expressed *NEUROG3* at different levels (Fig. 6c, Supplementary Fig. 7b). Some trunk cell markers, including *SOX9* and *ID3*, were expressed in EP early cells (Fig. 6c, Supplementary Fig. 7b), suggesting the role of EP early cells in the transitional state of trunk and EP cells. EP mid cells expressed both high levels of *NEUROG3* and *HES6* (Fig. 6c, Supplementary Fig. 7b). Some TFs, including *FEV*,

*HES6*, *PAX4*, and *NKX6-1*, were highly expressed in EP late cells (Fig. 6c, Supplementary Fig. 7b). EP alpha and EP beta cells still expressed low levels of *NEUROG3* and exhibited their own differentiation potential. EP alpha cells expressed *ARX*, *FEV* and *ISL1* but not *GCG*. EP beta cells expressed high levels of *NKX6-1*, *FEV*, and *PAX4* and low levels of *INS*, with no expression of mature beta cell marker genes, such as *MAFA* (Fig. 6c, Supplementary Fig. 7b). A recent study identified four subtypes of EP cells in PCW 9 to 19[28]. We compared our dataset with their dataset and found that most cell types could be matched (Supplementary Fig. 7c, d). Our newly defined EP alpha and EP beta cells mostly corresponded to their alpha/PP-Pro cells and EP4 cells (Supplementary Fig. 7c, d). Next, we focused on the TFs in endocrine cells and evaluated their activity by pySCENIC. Similar to the case for DEGs, the regulon (a

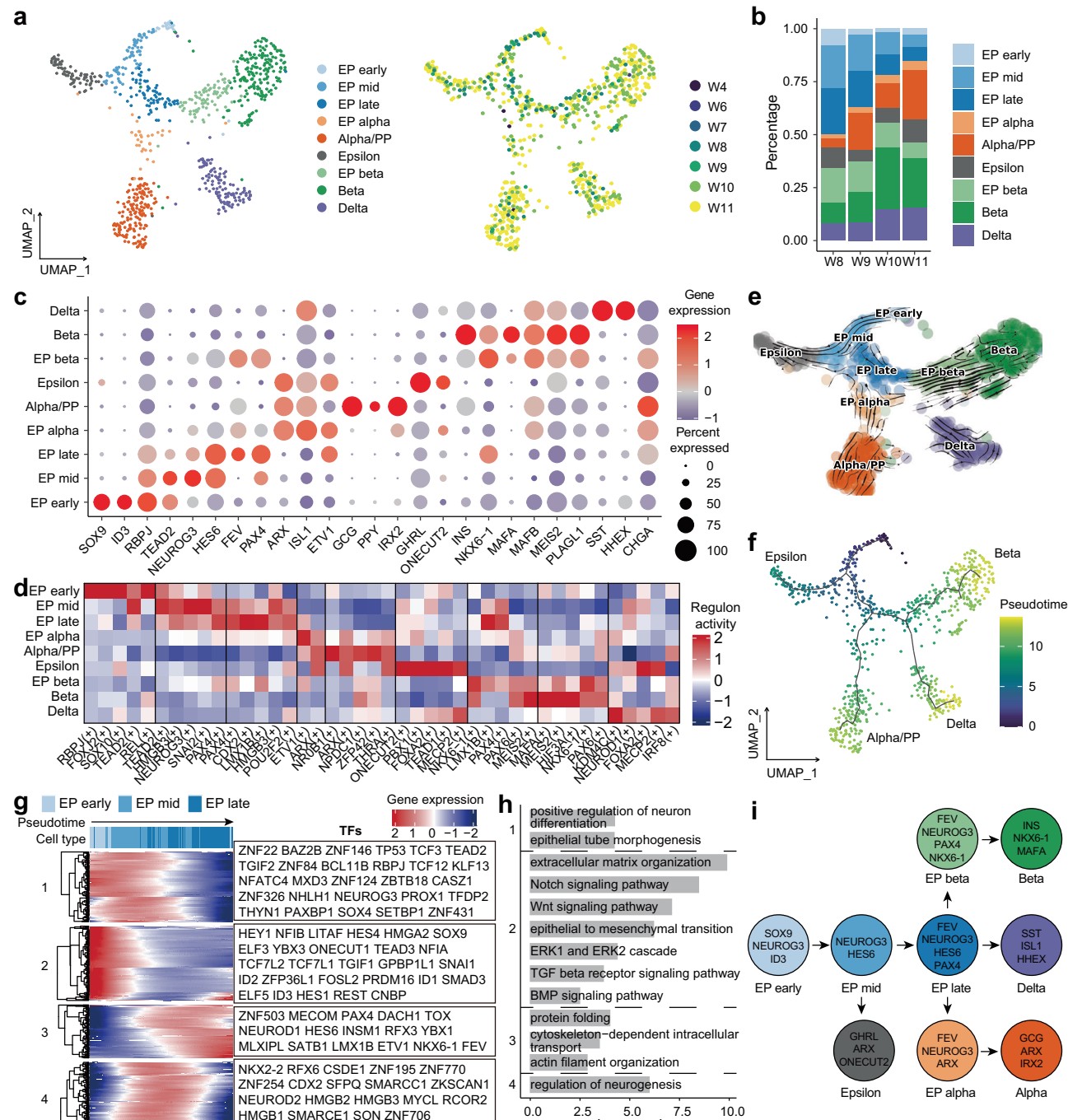

**Fig. 6 | Transcriptional heterogeneity of EPs and endocrine cells in the developing pancreas. a** UMAP plot of all single cells colored by cell type and time point in developing pancreatic endocrine cells. **b** Bar plot showing the percentage of each cell type in developing pancreatic endocrine cells. **c** Dot plot showing key marker gene expression in each cell type of developing pancreatic endocrine cells. **d** Heatmap showing the scaled activity of the top 5 highly scored regulons in each cell type of developing pancreatic endocrine cells. **e** RNA velocity plot showing the developmental trajectory of developing pancreatic endocrine cells. **f** UMAP plot showing the developmental trajectories of developing pancreatic endocrine cells. **g** Heatmap showing gene expression along the developmental trajectory of EP early to EP late cells. TFs are labeled on the right. **h** Selected enriched GO terms of genes in Fig. 6g. *P*-values were calculated by using *enrichGO* function from R package clusterProfiler with one-sided hypergeometric test. **i** Model of EP and endocrine cell differentiation. See also Supplementary Fig. 7 and Supplementary Data 5, 6.

given TF and its direct gene targets) activities were much different in each cluster (Fig. 6d, Supplementary Data 6). We highlighted the top 5 scoring regulons in each cluster and found that the ARX and ETV1 regulons scored highly in EP alpha cells and that the NKX6-1 and LMX1B regulons scored highly in EP beta cells (Fig. 6d).

Almost all EPs and endocrine cells were generated after PCW 8 (Fig. 6a, Supplementary Fig. 7a). The percentage of EP cells gradually

decreased, and that of endocrine cells increased between PCW 8 and 11 (Fig. 6b, Supplementary Fig. 7a). We performed RNA velocity and developmental trajectory analysis of pancreatic endocrine cells (Fig. 6e, f). EP early cells were considered the starting point of all endocrine cells. The epsilon cells were one branch from EP mid cells. Then, EP late cells differentiated first into EP alpha cells and then into delta and EP beta cells. Finally, EP alpha and EP beta cells only

generated alpha or beta cells (Fig. 6e, f). Next, we analyzed transcriptional dynamics according to their pseudotime. We divided the DEGs of EP early, EP mid, and EP late cells into four clusters along the pseudotime axis (Fig. 6g). Cluster-1 genes, including *TEAD2*, *SOX4*, and *NEUROG3*, were upregulated in EP early and EP mid cells (Fig. 6g). A GO analysis indicated that these genes participated in epithelial tube morphogenesis and neuron differentiation (Fig. 6h). Cluster-2 genes, such as *SOX9*, were highly expressed in EP early cells, and the GO terms enriched among these genes indicated that Notch, Wnt, BMP, TGF-β and extracellular signal-regulated kinase (ERK) pathways were down-regulated during EP development. The epithelial-mesenchymal transition (EMT) factor *SNAI1* and other related genes were also upregulated in EP early cells (Fig. 6g, h). The EMT process is believed to help EP cells delaminate into the mesenchyme from the trunk region in mice[56]. Cluster-3 genes, such as *PAX4*, *FEV*, *RFX3*, *NEUROD1*, *NKX6-1* and *INSM1*, were upregulated in EP late cells and associated with cytoskeletal organization for vesicle transport and protein folding (Fig. 6g, h). Cluster-4 genes associated with neurogenesis, including *NKX2-2*, *RFX6*, *NEUROD2* and *CDX2*, were upregulated in EP mid cells (Fig. 6g, h).

Alpha and beta cells constitute ~90% of the pancreatic endocrine cells in adults[57,58]. In the developing mouse pancreas, alpha cells emerge as early as E9.5 and precede beta cells. In the human pancreas, we found that *INS* expression starting from PCW 8 occurred earlier than *GCG* expression from PCW 9 (Supplementary Fig. 7f). This result indicated that the generation of insulin-expressing cells occurred prior to that of glucagon-expressing cells in human pancreas development and is consistent with previous findings[40,59]. To further evaluate alpha and beta cell differentiation, we focused on the branching of EP alpha and EP beta cells (Supplementary Fig. 7g). Insulin secretion, peptide transport, glucose homeostasis, and negative regulation of Notch signaling pathway-related genes were gradually upregulated in the beta branch (Supplementary Fig. 7g, h). Oxidative phosphorylation-, glucose homeostasis-, and peptide secretion-related genes were expressed at increasing levels in the alpha cell branch (Supplementary Fig. 7g, h). Taken together, our data depicted the differentiation trajectories of EPs and endocrine cells during the embryonic development of the human pancreas (Fig. 6i).

### Comparison of the embryonic pancreatic epithelium between humans and mice

To identify the molecular features of major pancreatic epithelial cell types between humans and mice during embryo development, we compared our pancreatic dataset with two published datasets for mouse embryonic pancreas from E9.5 to E17.5, the stage corresponding to human PCW 4 to 11[27,28]. We separately integrated the nonendocrine and endocrine datasets to make the results clearer (Fig. 7a, b). Our newly defined human dorsal MP cells were close to MP-early cells in mice (Fig. 7a, c). Some TFs, including *GATA5*, *RFX6*, *NKX6-2* and *NEUROG3*, were only expressed in mouse MP cells, while other TFs, including *NR2F1*, *TBX3* and *ID4*, were only expressed in human dorsal or ventral MP cells (Fig. 7d). The human early tip and early trunk cells that we identified were closer to a transition state between MP-late and tip or trunk cells in mice (Fig. 7a, c). This finding indicates that early tip-trunk differentiation in humans is milder than that in mice. We also identified some TFs that were differentially expressed between humans and mice (Fig. 7d); for example, *NR4A1*, *MAFF* and *EPAS1* were only expressed in human tip lineage cells, and *HES4*, *ASCL2* and *ID4* were only expressed in human trunk lineage cells (Fig. 7d). Regarding endocrine cells, human EP cells showed similar heterogeneity to mouse EP cells (Fig. 7b, c). Some TFs, such as *ZBTB18* and *ZNF503*, were only expressed in human EP cells (Fig. 7e). Epsilon cells had the potential to generate alpha and PP cells in mice, which was not observed in the human dataset, and consistent with previous results[28]. Alpha/PP cells still shared similar expression patterns with epsilon cells, for some TFs, such as *ARX*, *ETV1* and *ISL1* (Fig. 6c). *SOX4*, *SOX6*,

*LRRFIP1* and *ZFHX3* were highly expressed in human epsilon and alpha/PP cells (Fig. 7e). While *MAFA* was highly expressed in human beta cells, other functional maturation genes, such as *SLC2A2*, *IAPP* and *G6PC2*, were highly expressed in mouse beta cells (Fig. 7e). Additionally, TFs such as *PLAGL1*, *ASCL2*, *MNX1* and *SAMD11* were upregulated in human beta cells (Fig. 7e). Taken together, the results showed that although the developmental trajectories are conserved between humans and mice, a few distinct features, such as different gene expression patterns, were revealed between the two species.

## Discussion

In this study, we present an extensive analysis of the single-cell transcriptomic and chromatin accessibility profiles of human embryonic pancreas samples from the first trimester. The pancreas arises from both the dorsal and ventral endoderm domains, but it is not clear how the two parts contribute to the pancreas development. We first identified dorsal and ventral MP cells in humans and identified two new marker genes of these cells, *NR2F1* (dorsal) and *TBX3* (ventral). These results are similar to those of previous studies involving laser capture and deep sequencing in the human early dorsal pancreas[60]. In contrast to mouse pancreas development, there is only a single phase of *NEUROG3* expression and endocrine differentiation after PCW 8 in the human pancreas[13,26]. It might be that TFs related to endocrinogenesis, such as *RFX6* and *NEUROG3*, were expressed in mouse MP-early cells[27] but were not detected in our MP cells (Fig. 7d). The early dorsal pancreas and ventral pancreas receive different signals from adjacent tissues and may have distinct differentiation abilities[14,42]. Our data showed that the Wnt signaling pathway was more involved in dorsal MP cell development. A study in transgene-labeled mice showed that more endocrine cells were generated in the dorsal pancreas[61]. No significant differences in differentiation potentials between dorsal and ventral MP cells were observed in our data. In addition, these published scRNA-seq datasets also identified intermediate progenitors in the mouse ventral domain, which could generate hepatoblasts, EHBD cells, and pancreatic progenitors[61,62]. We only identified PB progenitors that could differentiate into ventral MP and EHBD cells in our dataset. Our earliest samples were from the CS13 stage, and intermediate progenitors that could also generate hepatoblasts might exist before CS13 in humans. *ISL1* was only expressed in the ventral foregut domain and not in the dorsal foregut domain in human CS10 and CS11 embryos[63]. Our identified PB progenitors showed high expression of *ISL1*, suggesting that PB progenitors exist only in the ventral foregut domain. In contrast, dorsal cell heterogeneity is less distinct than ventral cell heterogeneity, and no similar progenitor cell type has been identified for dorsal MP cells, suggesting that the dorsal foregut domain only generates the dorsal pancreas.

Tip-trunk differentiation is an important event in the development of the pancreas, and corresponding regulatory mechanism needs to be further investigated. The early tip and early trunk cells included in our dataset were not identified in a previous mouse dataset. They showed similar gene expression patterns and low expression of tip and trunk marker genes. We assume that early tip-trunk differentiation is less obvious in the early stage in humans than in mice. While PTF1A and NKX6-1 are considered the key TFs involved in tip and trunk differentiation, respectively, in mice[19]. Human tip cells still expressed *NKX6-1* (Supplementary Fig. 2d). We also found that Notch signaling and MAPK signaling from mesenchymal cells promoted trunk identity and HGF signaling for tip identity in humans. Benefiting from scATAC-seq data, we constructed GRNs of tip and trunk cells to identify important lineage-specific TFs. TFs downstream of Notch signaling, such as *HES1*, *HES4* and *HEY1*, drive MP cells to become ductal lineage cells, while TFs such as *PTF1A*, *XBP1* and *EPAS1* drive MP cells to become acinar lineage cells. The newly identified roles of TFs such as *GLIS3*, *NFIB* and *EHF* in ductal lineage cell differentiation and *EPAS1* in acinar lineage cell differentiation require further investigation. Notably, Glis3 has been

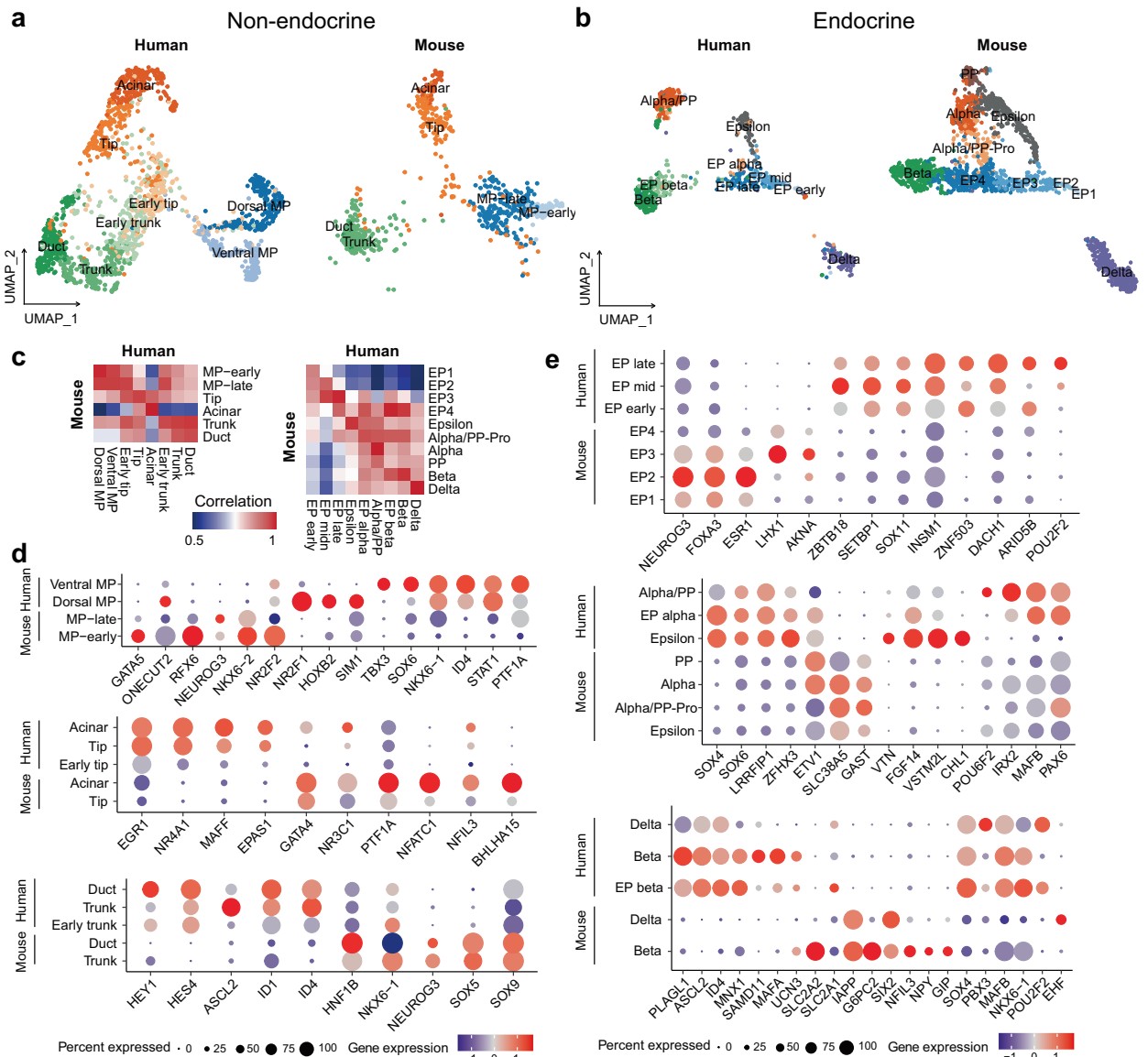

**Fig. 7 | Comparison of embryonic pancreatic epithelium between humans and mice. a** UMAP plot showing integration of our nonendocrine dataset with another mouse nonendocrine dataset. **b** UMAP plot showing integration of our endocrine dataset with another mouse endocrine dataset. **c** Heatmap showing the correlation of gene expression in different cell types between humans and mice. Pearson's correlation coefficient was calculated by average expression value of each gene in each sample. **d** Dot plot showing the expression of some different genes in none-ndocrine cells between humans and mice. **e** Dot plot showing the expression of some different genes in endocrine cells between humans and mice.

reported to regulate beta cell and duct cell development in mice[64], while our data imply that it is involved mainly in duct development in humans. Likewise, Epas1 has been reported to regulate beta cell and acinar cell development in mice, while its expression is confined to developing acinar cells in humans. Hence, their roles in human pancreas development need to be further studied in experiments with human developmental models such as organoids, rather than mouse embryos.

How EP cells differentiate from trunk cells is also poorly understood. Our data are consistent with previous studies showing that inhibiting the Notch, TGFβ, Wnt, BMP and ERK signaling pathways can promote EP cell differentiation from trunk cells in mice[27,43,65–68]. These conclusions have already been applied in the generation of endocrine cells from hPSCs in vitro[3,4,8,66,69,70]. Additionally, we found that epigenetic regulators, such as the histone deacetylase *HDAC2* and its components, were highly expressed in EP cells and might take part in EP cell differentiation. However, some previous studies showed that

HDAC inhibitor treatment of the embryonic rodent pancreas increased the number of EP cells[71,72]. The function of HDAC in EP cell differentiation in the human pancreas still needs to be validated. Moreover, in a combined analysis with scATAC-seq data, we identified TFs in trunk cells governing cell fate choice in endocrinogenesis; these TFs included *YBX1*, which upregulated *INSM1*, a reported important regulator in mouse islet development[52,53], to favor endocrine lineage differentiation. Our data suggested that *ID4*, which upregulated *HES4* in trunk cells, could suppress endocrine differentiation and favor ductal lineage development. Lineage tracing studies in mouse embryonic pancreas or human stem cell-derived organoids may further corroborate these discoveries.

Endocrineogenesis is the most attractive part of pancreas development. EP cell differentiation is a complex process and involves multiple intermediate cell states. We identified five EP subclusters with distinct differentiation potentials. Our data are largely consistent with a previous human model in which EP cells were clustered into four

subgroups[28]. We detected the same number of genes included in their mSTRT-seq dataset, while more EP cells were detected in our dataset (Supplementary Fig. 7e). In our model, epsilon cells were differentiated from EP mid cells corresponding to the EP2 and partial EP3 cells in the previous study[28]. EP late cells, which mainly related to most EP3 cells and some EP4 cells, showed multipotential to generate alpha, delta and beta cells. Moreover, we defined EP alpha cells as those corresponding to alpha/PP-Pro cells and EP beta cells corresponding to most EP4 cells. Compared with mouse endocrinogenesis, alpha and beta cell differentiation in humans takes more time. Epsilon cells generating alpha and PP cells and beta cell maturation may occur in the second and third trimesters. In addition, our data showed the *MAFA* expression in human fetal beta cells, consistent with previous studies showing that *MAFA* expression is low in the human developing pancreatic epithelium from PCW 9 onward[73,74]. MAFA is important for beta cell maturation, and its expression is increased in adult beta cells. However, nuclear MAFA protein did not appear until PCW 21[59,74], indicating that MAFA was located in the cytoplasm of early fetal beta cells and may not play a regulatory role as a transcription factor at this time point.

Taken together, our work provides a valuable resource for transcriptional dynamics in early human pancreas development as well as a blueprint for generating pancreatic cells in vitro.

## Methods

### Sample collection

The research complies with all relevant ethical regulations and guidelines. With approval from the Ethics Committee of The First Affiliated Hospital of Hainan Medical University (certificate #201901) and informed consent from the patients taking voluntary abortions, we acquired pancreas from human embryos in PCW 4-11.

The PCW of the embryos was determined by combining gestational age information, ultrasound assessments and anatomical features of the embryos according to the guidelines (Supplementary Data 1)[75–77]. In PCW 4-6, the GI tract part beneath the stomach and above the duodenum was dissected to include both the dorsal and ventral pancreas. In PCW 7-11, when the dorsal and ventral pancreas merged and the pancreas became distinct, the developing pancreas was separated, and as much of the surrounding mesenchyme was removed as much as possible. The dissected tissues were processed as appropriate for the subsequent experiments.

### scRNA-seq library construction and sequencing

For scRNA-seq library construction, the samples were washed with DPBS (Gibco), cut into millimeter pieces if necessary and digested in 0.25% trypsin (Gibco) at 37 °C for 5-10 min with regular inspection of the digestion status. Ten percent FBS (Gibco) was used to terminate the digestion once large tissue pieces were no longer visible, and the suspension was pipetted several times to further dissociate cell clumps. After washing with DPBS, the cell suspensions were filtered through 40-μm strainers (Bel-Art) to remove remaining cell clumps, followed by trypan blue (Sigma–Aldrich) staining for live cell counting. Cell counting was performed manually, and samples with a minimal cell viability of 90% were used for scRNA-seq with a target recovery of 5000-8000 cells. Library construction was performed using Chromium Next GEM Single Cell 3′ Reagent Kits v3.1 (10x Genomics) according to the manufacturer's instructions. The library was processed on the Illumina NovaSeq 6000 platform for sequencing with 150 bp paired-end reads.

### scATAC-seq library construction and sequencing

For scATAC-seq library construction, the samples were cryo-preserved with CryostorCS10 (STEMCELL). The nuclei were extracted according to the 10x Genomics user guide. Cryopreserved samples were recovered in prewarmed media (RPMI 1640 + 10% FBS) and then subjected to centrifugation at 300×g for 5 min at 4 °C. After DPBS washing, the cells were lysed in chilled lysis buffer (10 mM pH 7.4 Tris-HCl, 10 mM NaCl, 3 mM $MgCl_2$, 0.1% Tween-20, 0.1% Nonidet P40 Substitute, 0.01% digitonin, 1% BSA) with gentle pipette mixing and incubation on ice for 3 min. After washing with chilled wash buffer (10 mM pH 7.4 Tris-HCl, 10 mM NaCl, 3 mM $MgCl_2$, 0.1% Tween-20, and 1% BSA), the nuclei were resuspended in chilled diluted nuclei buffer (10x Genomics) and filtered through 40-μm strainers. The quality and quantity of nuclei were manually examined by trypan blue staining. Nuclei suspensions with a target recovery of 3000-6000 nuclei were subjected to Chromium Next GEM Single Cell ATAC Reagent Kits v1.1 (10x Genomics) according to the manufacturer's instructions. The library was processed on the Illumina NovaSeq6000 platform for sequencing with 50 bp paired-end reads.

### Cryosectioning and immunohistochemistry

For immunohistochemistry analysis, the samples were fixed in 4% paraformaldehyde (PFA, Sigma–Aldrich) for 24 h, embedded in optimum cutting temperature compound (OCT compound, Leica), quickly frozen in liquid nitrogen and stored at -80 °C. Cryosections were sectioned every 10 μm at −20 °C. Tissue sections were sequentially treated with 0.5% Triton X-100 for 10 min at room temperature, 5% BSA for 1 h at room temperature, diluted primary antibody solution at 4 °C overnight and diluted secondary antibody solution for 1 h at room temperature, with PBS wash between each step. Sections were incubated with primary or secondary antibodies at the following dilutions: Mouse anti-DCDC2 (C-4) (1:50, Santa Cruz, sc-166051), Mouse anti-ASCL2 (7E2) (1:100, Millipore, MAB4418), Rabbit anti-PDX1 (EPR22002) (1:100, Abcam, ab219207), Rabbit anti-HES4 (1:100, Invitrogen, PA5-84551), Goat anti-Mouse IgG (H + L) Cross-Adsorbed Secondary Antibody, Alexa Fluor 488 (1:400, Invitrogen, A-11001), Donkey anti-Rabbit IgG (H + L) Highly Cross-Adsorbed Secondary Antibody, Alexa Fluor 568 (1: 400, Invitrogen, A-10042). Immunofluorescence images were taken with an Olympus FV3000 confocal microscope.

### scRNA-seq processing

The sequencing output files were processed with Cell Ranger 4.0.0 with default parameters, aligning reads to the GRCh38 (hg38) reference genome. SoupX[78] was deployed to remove ambient RNA contamination from each count matrix, and the data were then inputted to the Seurat workspace[79]. Cells with UMI counts between 4000 and 50,000, gene numbers between 500-8000 and mitochondrial count percentages <15% were used for further analysis.

The Seurat objects from the PCW 4-6 and PCW 7-11 samples were simply merged together, respectively. The two datasets were normalized by the total counts per cell. The 3000 most variable genes were selected. Cell cycle scores were calculated by the CellCycleScoring function and regressed out together with sequencing depth and batch. Principal component analysis (PCA) was performed with a z score matrix, followed by UMAP and Louvain clustering applied with the top 50 PCs and a resolution of 1. Cells coexpressing markers of more than one cell class were treated as doublets and removed from the datasets. The markers used for cell type annotation are listed in Supplementary Fig. 1e, g.

Then, pancreatic epithelium from different datasets (PCW 4-6 or PCW 7-11) were extracted and merged together, followed by repeating the analysis process described above with parameters modified. Briefly, UMAP and Louvain clustering were applied with the top 40 PCs and a resolution of 2.5. The markers used for cell type annotation are listed in Supplementary Fig. 2d.

For the PCW 4-5 epithelial cells shown in Fig. 2, epithelial cells in PCW 4-5 samples were extracted, followed by repeating the analysis process demonstrated above with some procedures modified. Briefly, 2000 highly variable genes were selected, and cell cycle scores and sequencing depth were regressed out. After PCA, we used the package Harmony to perform batch effect correction. Next, UMAP and Louvain

 

clustering were applied with the top 30 Harmony reduction and a resolution of 0.7. The markers used for cell type annotation are listed in Fig. 2e.

For the pancreatic endocrine cell dataset, EP, alpha, beta, delta and epsilon cells were extracted from the pancreatic epithelial cell dataset, followed by repeating the analysis process described above with some parameters modified. Briefly, 2000 highly variable genes were selected, and cell cycle scores and sequencing depth were regressed out. After PCA, we used the package Harmony[80] to perform batch effect correction. Next, UMAP and Louvain clustering were applied with the top 35 Harmony reduction and a resolution of 2. The markers used for cell type annotation are listed in Fig. 6c.

DEGs for each cluster in all datasets were listed by the *FindAllMarkers* function with the default parameters, and only genes for *p*-value adjusted <0.05 were selected.

## Pseudotime and trajectory analysis
Pseudotime analysis was completed with the Monocle3 package[37]. The gene expression matrix and UMAP layout were imported to the Cell-DataSet object. The *learn_graph* functions were used to build trajectories with the parameters *minimal_branch_len = 4*. The *order_cells* function was employed to measure the pseudotime of each cell, and cells were manually selected as root nodes of the trajectory graph. For gene expression dynamics visualization, we extracted a DEG-related expression matrix for cell types of interest and arranged the cells by pseudotime. Then, *smooth.spline* was applied to fit the gene expression. The package ComplexHeatmap[81] was applied to visualize the gene expression and cluster genes by the k-means method.

## RNA velocity analysis
Loom files containing unspliced and spliced reads were generated from *velocyto.py* for downstream analysis with default parameters[38]. Then, the python package scVelo[39] was applied to estimate RNA velocity with a dynamic model by default parameters. Later, RNA velocities were projected into UMAP embedding and visualized by cell types.

## GO analysis
GO analysis was performed with the package clusterProfiler[82]. Only biological process GO terms with a *p*-value < 0.05 were selected. The results were further visualized with the ggplot2 package.

## Cell–cell interaction analysis
Cell–cell interaction analysis was performed with the package CellChat[83]. The *CellChatDB.human* database used in our analysis contains secreted signaling, ECM-receptor and cell–cell contact ligand–receptor (LR) interactions. The Seurat object was imported into Cell-Chat object and processed according to the guidelines. The chord diagram of the signaling pathway and LR interactions of interest was fulfilled by the *netVisual_chord_gene* function. Only LR interactions with a *p*-value < 0.05 were selected.

## Identification of regulons of endocrine cells
The count matrix of endocrine cells was input into the pySCENIC workflow with default parameters[84,85]. First, the gene coexpression network was constructed by the *grn* step. Second, each TF-target module was pruned with a previously known regulatory motif in the *cisTarget* database by the *ctx* step. Then, the AUC score for each regulon was calculated at the single-cell level by *aucell* step. Finally, differentially activated regulons in each cluster were identified by the Wilcoxon test, and only *p*-values adjusted <0.05 were selected.

## scATAC-seq processing
The sequence output files were processed with Cell Ranger ATAC 2.0.0 with default parameters, aligning reads to the GRCh38 (hg38)

reference genome and identifying transposase cut sites. The Signac package[86] was used to preprocess scATAC-seq data. Cells with a TSS enrichment score between 4 to 20, nucleosome signal score <4 and blacklist ratio <0.02 and total <50,000 were selected for further analysis.

Genomes were segmented into 2.5 kb windows, and windows containing blacklist regions were removed. Count matrices of each sample were constructed by the *FeatureMatrix* function, binarized and merged together prior to term frequency inverse document frequency (TF-IDF) normalization (method 3 of function *RunTFIDF*). The 25000 most accessible windows were subjected to singular value decomposition (SVD), and the *RunHarmony* function was used to remove batch effects with 2-30 LSI components. The 2-30 Harmony components were then used for UMAP (n.neighbors = 50 L, min.dist = 0.5) and Louvain clustering (resolution = 0.8). The gene activity matrix was generated using the GeneActivity function. Cells coexpressing markers of more than one cell classes or expressing none of the markers of interested cell classes were removed from the datasets. The markers used for cell class annotation are listed in Supplementary Fig. 4c.

Epithelial cells were extracted from the whole dataset and then subjected to peak calling. The method described in a previous study[87] was used to generate 500 bp peaks. Briefly, the *CallPeaks* function in Signac was used to call peaks for each Seurat cluster with additional parameters "*--call-summits --nolambda --qval 5e-2 --keep-dup all*" passed to MACS2[88]. After extending peak summits by 250 bp on both sides, any peaks aligning to blacklist regions were removed. The 200000 most significant peaks were selected after the iterative selection described in a previous study[87]. The peak matrices were generated by applying the *FeatureMatrix* function to the selected peaks, binarizing the peaks and subjecting them to the clustering process described above. Due to the inherent sparsity of scATAC-seq data, we applied a broader strategy in annotating epithelial cells in scATAC-seq by ignoring differences between different stages of progenitors (e.g., early tip and tip cells were both annotated as tip cells) and focusing on differences between lineages. The markers used for cell type annotation are listed in Supplementary Fig. 4d. Differentially accessible peaks for each cell type were calculated by the *FindAllMarkers* function with the parameter *min.pct* set to 0.01 and *logfc.threshold* set to 0.15. Motif information was added by the *AddMotifs* function with the filtered *cisBP* database (human_pwms_v2 from the chromVARmotifs package)[89]. Motif enrichment for each cell type was calculated by performing *FindMotifs* on differentially accessible peaks for each cell type.

## Joint analysis of scRNA-seq and scATAC-seq
The ArchR package[90] was employed in integrated analysis, and the established LSI coordinates, Harmony coordinates, UMAP coordinates, cell type annotation and gene activity matrix described above were inputted to ArchR. To integrate time-matched scRNA-seq and scATAC-seq data, data for PCW 8-11 epithelial cells were extracted from scRNA-seq datasets. scRNA-seq and scATAC-seq were integrated by unconstrained integration using the *addGeneIntegrationMatrix* function on the gene activity matrix imported from Signac. Peak-to-gene links were then added using the *addPeak2GeneLinks* function.

## GRN analysis
GRNs were constructed according to IReNA2 with slight modifications[49]. Briefly, TF footprints were calculated by TOBIAS[91]. Motif enrichment and peak-to-gene links were calculated as described above. TF footprint and motif enrichment were used to assess TF activity, while peak-to-gene links and gene coexpression relationships were exploited to predict targets. Expression correlations were used to screen predicted TF-target pairs. Default parameters described in the original paper were used[49].

For acinar lineage GRN analysis, cell types involved in the acinar lineage, namely, tip and acinar cells for scATAC-seq and early tip, tip and acinar cells for scRNA-seq, were treated all as acinar lineage cells, and cell type distinctions within this group were ignored. For ductal lineage GRN analysis, a similar strategy was employed, ignoring differences between trunk and duct cells for scATAC-seq and early trunk, trunk and duct cells for scRNA-seq.

For the analysis of GRNs regulating the differentiation of trunk cells to duct or endocrine progenitors, GRNs of trunk, duct and endocrine progenitors were constructed separately. Then, TF-target pairs regulating bidirectional differentiation were selected based on two criteria (exemplified by trunk-endocrine transition): (a) the targets in trunk cells were TFs in endocrine cells but not in duct cells; (b) the TF-target pairs were shared between trunk cells and endocrine cells but not duct cells, and for trunk-duct transitions, the opposite criteria were applied.

### Comparison with published another dataset for endocrine cells

We used the mSTRT-seq dataset from OMIX236 for human embryonic pancreas in PCW 9 to 19 to compare with our endocrine cell dataset[28]. Only endocrine cells were extracted. Then, the standard workflow of Seurat was applied to integrate the two datasets by canonical correlation analysis[92]. The two datasets were separately visualized in the same UMAP space. Cell type identity transfer was performed using the *FindTransferAnchors* and *TransferData* functions with default parameters and visualized by an alluvial plot in the ggplot2 package.

### Comparison across species between humans and mice

Cross-species comparison was based on genes with a 1:1 ortholog between humans and mice by the BioMart from Ensembl genome annotation system (http://www.ensembl.org/index.html), and the mouse gene names were converted to human gene names. To overcome biases due to unbalanced cell numbers, we sampled 200 cells per cell type for our human nonendocrine cells to form a balanced dataset. This dataset was integrated with the Smart-seq2 dataset from GSE115931[27]. Our endocrine cell dataset was integrated with the Smart-seq2 dataset from GSE139627[28]. Only cells from E9.5 to E17.5 were applied to integrate with our data. Then, the standard workflow of Seurat was applied to integrate two datasets by canonical correlation analysis. The two datasets were separately visualized in the same UMAP space. DEGs between humans and mice across the same cell types were calculated by the *FindMarkers* function.

### Statistics and reproducibility

Statistical parameters are reported in the respective figures and figure legends.

### Reporting summary

Further information on research design is available in the Nature Portfolio Reporting Summary linked to this article.

### Data availability

The raw sequencing data generated in this study have been deposited in the Genome Sequence Archive for Human (GSA-Human) under accession code HRA002757. The data in GSA are available under restricted access for privacy protection, access can be obtained by contacting Tao Xu (xutao@ibp.ac.cn). China's Ministry of Science and Technology has approved the export of raw sequencing data (approval # 2023BAT1021). The processed gene expression matrix for scRNA-seq and Tn5 fragment files and filtered peak-barcode matrix for scATAC-seq data are available at OMIX database under accession code OMIX001616. Source data are provided with this paper. Other published datasets we used in this study could be obtained from GSE115931, GSE139627 and OMIX236. Source data are provided with this paper.

### Code availability

The code for scRNA-seq and scATAC-seq analysis during this study is available at Github: https://github.com/zhuoma888/fetal_pancreas.

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

## Acknowledgements

This work was supported by grants from the Ministry of Science and Technology of the People's Republic of China (2021YFA1300301, 2018YFA0507101), the Beijing Natural Science Foundation (5212016), the National Natural Research Foundation of China and the Swedish Foundation for International Cooperation in Research and Higher Education (NSFC-STINT) Cooperation Project (32011530118), the Academic Promotion Program of Shandong First Medical University (922-001003130RC), the SciLifeLab & Wallenberg Data Driven Life Science Program (KAW 2020.0239), the Major Science and Technology Program of Hainan Province (ZDKJ2017007), the National Natural Science Foundation of China (81960283), Hainan Provincial Science and Technology Program for Clinical Medical Research Center (LCYX202102), the Hainan Province Clinical Medical Center, the Innovation Platform for Academicians of Hainan Province. We are grateful to Dr. Zhen Fan from Center for High Throughout Sequencing for the technical support with the preparation of single-cell samples and the construction of the scRNA-seq and scATAC-seq library. We thank Dr. Yan Teng and her group from Center for Biological imaging for helping with the bioimaging. We would also like to thank the Guangdong-Hong Kong-Macao Greater Bay Area Center of Bioinformation (GCBI) for the support of computing resources.

## Author contributions

E.S. and T.X. conceived the project and designed the experiments. Z.M., X.Z., E.S., H.Y. and Y.M. collected the human embryo samples. Z.M. and X.Z. performed tissue processing, single-cell RNA-seq and single-cell ATAC-seq experiments. Z.M., W.Z. and X.C. analyzed the scRNA-seq data. X.Z. and W.Z. analyzed the scATAC-seq data. X.C. contributed to data visualization. Z.M., X.Z., and Y.Z. performed immunostaining, imaging, and data analysis. Z.M. and X.Z., E.S., W.Z. and T.X. wrote and revised the manuscript. All authors contributed to the discussion and interpretation of the results.

## Competing interests

The authors declare no competing interests.
