## [Peer Review File · Nature Communications]

Deciphering early human pancreas development at the single-cell levelREVIEWER COMMENTS

Reviewer #1 (Remarks to the Author):

Nature Communications Ma et al.

Ma and colleagues single cell transcriptome and chromatin accessibility data for early fetal human pancreas development. The data appear of high quality, but – by its nature – the study is entirely descriptive, and recapitulates largely what is known from the mouse pancreas.

Specific comments:

Introduction:

Line 62: The enthusiasm should be tempered substantially. hPSC derived beta cells MAY be a therapeutic approach for T1D, certainly not for a long time for T2D. Also, the clinical trials have not been concluded yet, and safety and efficacy concerns have not been addressed. However, it is true that the methods developed to coax stem cells towards a beta like phenotype have been based on decades of work delineating mammalian pancreas development using mouse genetics. Key papers need to be cited to acknowledge this fact regarding the function of the FoxA, PDX1, NGN3, and MAFA factors at the minimum. Line 81, while it is true that subtle differences between mouse and human pancreas development exist, the fact remains that the in vitro differentiation protocols were derived based on mouse findings.

Results:

The manuscript is riddled with 'private' abbreviations such as "MP", PCW, EP, VP, DP etc. Please spell these out to increase readability.

The scATACseq data are not well integrated.

Discussion:

This could be shortened substantially, as it repeats a lot of information from Introduction and Results.

Reviewer #2 (Remarks to the Author):

Ma and colleagues present compelling data delineating the development of human pancreatic cell lineages, their interactions, and relationships, as well as the gene regulatory programs underlying their differentiation. The findings provide a comprehensive picture of how the early mid-foregut area cells differentiate into the various pancreatic and hepatic cells and how distinct progenitor populations give rise to developing endocrine cell populations. Several comments should be addressed.

The statement that ventral and dorsal MP cells express the common pancreas markers PDX1, PTF1A, SOX9, and NKX6.1 does not seem to be reflected by the data in Fig. 2B. There appears to be a clear reduction in the expression of these markers in dorsal MP cells, which would be a remarkable and unexpected finding. This difference in expression is not

apparent in the data presented in Fig. 2E. Showing actual expression data would help to understand how distinct the expression levels of these genes actually are.

The identification of PB progenitors marked by low expression of PDX1 is interesting, particularly because they seem to be related exclusively to ventral MP cells, but not dorsal MP cells. The authors should discuss if PB cells give rise to dorsal pancreas cells. If not, is there a similar progenitor cell type for dorsal MP cell populations?

Another confounding observation is that the authors observe clear interactions/cell communication between supporting non-epithelial and trunk/duct cells with the involvement of defined signaling pathways. In contrast, similar communication between mesenchymal and tip-acinar cells is not described in the manuscript. Is there any evidence for such interactions and if not, the authors should discuss the absence of these instructive/missive signals for tip-acinar cell differentiation.

Data in figure. 6C indicate that at least 50% of the embryonic beta cells express MAFA at a robust/high level. These data contrast with prior work that found increased expression of MAFA only after the first decade of life in humans (Arda et al, 2016). The authors should discuss this discrepancy. Also, what is the expression level of MAFB that was previously noted to be expressed before MAFA in developing pancreatic beta cells?

Prior studies had identified INSULIN-GLUCAGON double positive cells in human embryos. Does this cell population exist in the data sets analyzed here?

The Carnegie stages should be defined briefly and correlated with embryonic age.

Reviewer #1 (Remarks to the Author):

Nature Communications Ma et al.

Ma and colleagues single cell transcriptome and chromatin accessibility data for early fetal human pancreas development. The data appear of high quality, but – by its nature – the study is entirely descriptive, and recapitulates largely what is known from the mouse pancreas.

Specific comments:

Introduction:

Line 62: The enthusiasm should be tempered substantially. hPSC derived beta cells MAY be a therapeutic approach for T1D, certainly not for a long time for T2D. Also, the clinical trials have not been concluded yet, and safety and efficacy concerns have not been addressed. However, it is true that the methods developed to coax stem cells towards a beta like phenotype have been based on decades of work delineating mammalian pancreas development using mouse genetics. Key papers need to be cited to acknowledge this fact regarding the function of the FoxA, PDX1, NGN3, and MAFA factors at the minimum.

Response: We thank the reviewer for the comments. We have modified the description of hPSC-derived beta cells and related contents. We have also cited some key papers on the functions of key transcription factors, including *FoxA*, *PDX1*, *NGN3*, and *MAFA*, in the parts describing pancreas organogenesis, as suggested.

Line 81, while it is true that subtle differences between mouse and human pancreas development exist, the fact remains that the in vitro differentiation protocols were derived based on mouse findings.

Response: Thanks for pointing out this fact. We have added this information and modified the related contents.

Results:

The manuscript is riddled with 'private' abbreviations such as "MP", PCW, EP, VP, DP etc. Please spell these out to increase readability.

Response: Sorry for the misunderstanding. We avoided some uncommon abbreviations, including DP and VP, in the current manuscript. We also carefully read the revised manuscript and make sure that the full names were used when they first appeared in the paper according to the usual practice. Additionally, we summarized an abbreviation list as follows. We will add the list in the paper if it can be accepted for the requirement of *Nature Communications* format.

Abbreviation list : hPSC, human pluripotent stem cell; EHBD, extrahepatic bile duct; MP, multipotent progenitor; EP, endocrine progenitor; TF, transcription factor; scRNA-seq, single-cell RNA sequencing; scATAC-seq, single-cell assay for transposase accessible chromatin sequencing; PCW, post-conception week; GO, Gene Ontology; DEG, differentially expressed gene; PB, pancreato-biliary; GRN, gene regulatory network; CCA, canonical correlation analysis.

The scATACseq data are not well integrated.

Response: We thank the reviewer for the comment. To correct the batch effects in the scATAC-seq data, we used the harmony method, in which two dimensions in the harmony embeddings showed clear removal of batch differences (Figure R1a). The cell types in different post conception weeks (PCWs) showed that they had similar marker gene expression patterns, avoiding cell type confounding resulting from batch correction (Figure R1b). UMAP plots were revised to better visualize the cell type maturation during development. The new UMAP plot showed that the pancreas in PCW 8 and PCW 9 contained mainly immature progenitors, while mostly mature cell types existed in PCW 10 and PCW 11 (Figure R1c). We also performed canonical correlation analysis (CCA) on the scATAC-seq data. As shown in Figure R1d and R1e, the cell type annotations generated by CCA and harmony correction were consistent.

However, as mentioned in the limitations paragraph of the manuscript, only one

sample at each time point was analyzed in the scATAC-seq data due to human embryo scarcity. Future replicates at each time point may offer further guidance for integration to avoid overcorrection.

Figure R1. Integration analysis of scATAC-seq data.

- (a) Batch effects before (left) and after (right) harmony correction.
- (b) Dot plot showing marker gene expression in cell types in each post conception weeks.
- (c) Revised UMAP plot showing the integration of scATAC-seq using harmony.
- (d) UMAP plot showing the integration of scATAC-seq using canonical correlation analysis (CCA).
- (e) Heatmap comparing the cell type annotation results using harmony and CCA correction

Discussion:

This could be shortened substantially, as it repeats a lot of information from Introduction and Results.

Response: Thanks for the suggestion. We have deleted some repeated information and added some new information according to reviewer #2's suggestions.

Reviewer #2 (Remarks to the Author):

Ma and colleagues present compelling data delineating the development of human pancreatic cell lineages, their interactions, and relationships, as well as the gene regulatory programs underlying their differentiation. The findings provide a comprehensive picture of how the early mid-foregut area cells differentiate into the various pancreatic and hepatic cells and how distinct progenitor populations give rise to developing endocrine cell populations. Several comments should be addressed.

The statement that ventral and dorsal MP cells express the common pancreas markers PDX1, PTF1A, SOX9, and NKX6.1 does not seem to be reflected by the data in Fig. 2B. There appears to be a clear reduction in the expression of these markers in dorsal MP cells, which would be a remarkable and unexpected finding. This difference in expression is not apparent in the data presented in Fig. 2E. Showing actual expression data would help to understand how distinct the expression levels of these genes actually are.

Response: We thank the reviewer for the constructive suggestions. We have modified Fig. 2c and used box plots to show the differential expression of these genes (Figure R2). We have also deleted some genes and modified the description of this figure due to space limitation. As for the recognized pancreatic markers *PDX1*, *PTF1A*, *SOX9*, and *NKX6.1*, only *PTF1A* was differentially expressed gene between dorsal MP and ventral MP cells.

Figure R2. Different expression of key genes in dorsal and ventral MP cells.

Box plots showing the expression of key genes in dorsal and ventral MP cells. The numbers above the box plots represent the p-values calculated using the Wilcoxon test.

The identification of PB progenitors marked by low expression of PDX1 is interesting, particularly because they seem to be related exclusively to ventral MP cells, but not dorsal MP cells. The authors should discuss if PB cells give rise to dorsal pancreas cells. If not, is there a similar progenitor cell type for dorsal MP cell populations?

Response: Thanks for the comment and question. The ventral pancreas, liver, gallbladder and extrahepatic bile ducts all originate from the ventral foregut domain. The dorsal pancreas originates from the dorsal foregut domain. The newly identified PB progenitors highly expressed the *ISL1* and *HHEX* genes. *ISL1* has been recently reported to be expressed only in the ventral foregut domain compared with the dorsal foregut domain in human CS10 and CS11 embryos¹. *Hhex* has been identified as essential for liver and bile duct development in mouse embryos²⁻⁴. Thus, we propose that PB progenitors exist only in the ventral foregut domain and cannot give rise to dorsal MP cells, while the dorsal foregut domain, corresponding to the ventral hepatopancreato-biliary system, develops only into the dorsal pancreas. In contrast, dorsal cell

heterogeneity is less distinct than ventral cell heterogeneity. No similar progenitor cell type has been identified for dorsal MP cells. We have added this information to the revised Discussion.

Another confounding observation is that the authors observe clear interactions/cell communication between supporting non-epithelial and trunk/duct cells with the involvement of defined signaling pathways. In contrast, similar communication between mesenchymal and tip-acinar cells is not described in the manuscript. Is there any evidence for such interactions and if not, the authors should discuss the absence of these instructive/missive signals for tip-acinar cell differentiation?

Response: We thank the reviewer for the constructive question and suggestion. We found that the HGF signaling pathway was important for acinar lineage cells (Figure R3). *MET* was specifically expressed in tip and acinar cells, and its ligand *HGF* was expressed in pericytes and mesothelial cells. We have modified Fig. 4g and Supplement Fig. 5 as well as the related text to discuss the HGF signaling in acinar lineage cells.

Figure R3. Cell-cell interaction between acinar lineage cells and supporting cells.

- (a) Network plot showing HGF signaling interactions between acinar, ductal lineage cells and supporting cells.
- (b) Dot plot showing the communication probability and p-value of selected interactions between acinar and ductal lineage cells and supporting cells.
- (c) Violin plot showing the expression of HGF signaling receptor MET in acinar and ductal lineage cells.
- (d) Violin plot showing the expression of HGF signaling ligand HGF in supporting cells.

Data in figure. 6C indicate that at least 50% of the embryonic beta cells express MAFA at a robust/high level. These data contrast with prior work that found increased expression of MAFA only after the first decade of life in humans (Arda et al, 2016). The authors should discuss this discrepancy. Also, what is the expression level of MAFB that was previously noted to be expressed before MAFA in developing pancreatic beta cells?

Response: Thanks for the question. MAFA and MAFB are both important for beta cell development in humans. Previous studies have observed *MAFA* expression in the human developing pancreatic epithelium in PCW 9 at a low level^{5, 6}. These data are consistent with our scRNA-seq data (Figure R4). With regard to *MAFA*, its expression is much higher in adult islets⁶. However, nuclear MAFA protein has not been demonstrated to exist until PCW 21^{5, 7}, indicating that MAFA is located in the cytoplasm in early fetal beta cells and may not play a regulatory role as a transcription factor at this timepoint. Then, *MAFA* expression increased and MAFA regulated beta cell maturation in third-trimester fetuses and neonatal islets. Similar to the situation in mice, *MAFB* expression occurs prior to *MAFA* expression in human developing beta cells. The difference is that adult human beta cells maintain *MAFB* expression along with *MAFA*. We have added a discussion about *MAFA* expression in human developing beta cells in the revised manuscript.

Figure R4. MAFA and MAFB expression in human developing pancreatic endocrine cells.

(a) Dot plot showing the expression of MAFA and MAFB in human developing pancreatic endocrine cells.

(b) Violin plots showing the expression of MAFA and MAFB in human developing pancreatic endocrine cells.

Prior studies had identified INSULIN-GLUCAGON double positive cells in human embryos. Does this cell population exist in the data sets analyzed here?

Response: As suggested, we identified some alpha/PP cells that coexpressed the *INS* and *GCG* genes in our dataset (Figure R5). These double-positive cells have an alpha cell fate, which is consistent with previous studies ⁵.

Figure R5. INS and GCG expression in human developing pancreatic endocrine cells.

(a) UMAP plot of all single cells colored by cell type and time point in developing pancreatic endocrine cells.

(b) Feature plot showing the coexpression levels of INS and GCG in developing pancreatic endocrine cells.

The Carnegie stages should be defined briefly and correlated with embryonic age.

Response: Thanks for the suggestion. We have summarized the embryo information, including their Carnegie stages, corresponding embryonic age and days post-conception, in Supplement Data 1.

Reference

1. Li, L.C. *et al.* Single-cell patterning and axis characterization in the murine and human definitive endoderm. *Cell Res* **31**, 326-344 (2021).
2. Keng, V.W. *et al.* Homeobox gene Hex is essential for onset of mouse embryonic liver development and differentiation of the monocyte lineage. *Biochem Biophys Res Commun* **276**, 1155-1161 (2000).
3. Hunter, M.P. *et al.* The homeobox gene Hhex is essential for proper hepatoblast differentiation and bile duct morphogenesis. *Dev Biol* **308**, 355-367 (2007).
4. Martinez Barbera, J.P. *et al.* The homeobox gene Hex is required in definitive endodermal tissues for normal forebrain, liver and thyroid formation. *Development* **127**, 2433-2445 (2000).
5. Riedel, M.J. *et al.* Immunohistochemical characterisation of cells co-producing insulin and glucagon in the developing human pancreas. *Diabetologia* **55**, 372-381 (2012).
6. Sarkar, S.A. *et al.* Global gene expression profiling and histochemical analysis of the developing human fetal pancreas. *Diabetologia* **51**, 285-297 (2008).
7. Jeon, J., Correa-Medina, M., Ricordi, C., Edlund, H. & Diez, J.A. Endocrine cell clustering during human pancreas development. *J Histochem Cytochem* **57**, 811-824 (2009).

REVIEWERS' COMMENTS

Reviewer #1 (Remarks to the Author):

The authors have addressed my prior comments in a satisfactory manner.

Reviewer #2 (Remarks to the Author):

The revised version of the manuscript addresses my concerns. I would recommend publication of the manuscript.

Reviewer #1 (Remarks to the Author):

The authors have addressed my prior comments in a satisfactory manner.

Response: Thanks.

Reviewer #2 (Remarks to the Author):

The revised version of the manuscript addresses my concerns. I would recommend publication of the manuscript.

The Carnegie stages should be defined briefly and correlated with embryonic age.

Response: Thanks.